# Identification of putative enhancer-like elements predicts regulatory networks active in planarian adult stem cells

**Jakke Neiro[1]\*, Divya Sridhar[1], Anish Dattani[2], Aziz Aboobaker[1]\***

[1]Department of Zoology, University of Oxford, Oxford, United Kingdom; [2]Living Systems Institute, University of Exeter, Exeter, United Kingdom

**Abstract** Planarians have become an established model system to study regeneration and stem cells, but the regulatory elements in the genome remain almost entirely undescribed. Here, by integrating epigenetic and expression data we use multiple sources of evidence to predict enhancer elements active in the adult stem cell populations that drive regeneration. We have used ChIP-seq data to identify genomic regions with histone modifications consistent with enhancer activity, and ATAC-seq data to identify accessible chromatin. Overlapping these signals allowed for the identification of a set of high-confidence candidate enhancers predicted to be active in planarian adult stem cells. These enhancers are enriched for predicted transcription factor (TF) binding sites for TFs and TF families expressed in planarian adult stem cells. Footprinting analyses provided further evidence that these potential TF binding sites are likely to be occupied in adult stem cells. We integrated these analyses to build testable hypotheses for the regulatory function of TFs in stem cells, both with respect to how pluripotency might be regulated, and to how lineage differentiation programs are controlled. We found that our predicted GRNs were independently supported by existing TF RNAi/RNA-seq datasets, providing further evidence that our work predicts active enhancers that regulate adult stem cells and regenerative mechanisms.

**\*For correspondence:**
jakke.neiro@queens.ox.ac.uk
(JN);
aziz.aboobaker@zoo.ox.ac.uk
(AA)

**Competing interest:** The authors declare that no competing interests exist.

## Editor's evaluation

The authors have generated a comprehensive list of transcription factors in the planarian *Schmidtea mediterranea*, providing insight into which factors are highly expressed in the stem cell compartment. The computational identification of transcription factors and putative enhancers will be helpful to researchers studying stem cell and regenerative biology using planarians and provides a large dataset that informs upon the evolution of regulatory sequences and transcription factor function.

## Introduction

The molecular and evolutionary mechanisms of regeneration remain underexplained compared to animal development. This can be attributed to the greater technical difficulty historically associated with studying the molecular mechanisms of adult biology compared with development. However, an ever-broadening repertoire of model organisms for regeneration, progress in understanding the variety of cellular and molecular mechanisms used across taxa, and advances in experimental tools are serving to close this gap. One area of slower progress has been our knowledge of the regulatory logic of regenerative mechanisms, with relatively few studies in highly regenerative models providing precise insight into the epigenetic regulation of regeneration. While transcription factors (TFs) have been assigned functions, and in some cases a list of likely targets through transcriptome analysis, regulatory logic has been studied in only a few cases (*Pascual-Carreras et al., 2020*; *Wang et al.,*

*2020*). This is particularly true in the highly regenerative planarian model system, where only a few studies credibly address epigenetic and regulatory mechanisms (*Duncan et al., 2015*; *Dattani et al., 2018*; *Mihaylova et al., 2018*; *Dattani et al., 2019*, *Pascual-Carreras et al., 2020*).

Enhancers are distal-acting elements that regulate transcription initiation when TFs attach to TF binding sites, containing conserved sequences. In a regenerative context, active enhancers have been identified at the tissue level in zebrafish, the African killifish *Nothobranchius furzeri,* and the fruit fly *Drosophila melanogaster*, and at the whole-body level in the acoel worm *Hofstenia miamia* and the cnidarian *Hydra vulgaris* (*Harris et al., 2016*, *Kang et al., 2016*; *Goldman et al., 2017*; *Gehrke et al., 2019*; *Murad et al., 2019*, *Yang and Kang, 2019*; *Wang et al., 2020*). Despite their importance as a model for understanding regeneration, there are only a few putative enhancers identified or implicated in planarians (*Pascual-Carreras et al., 2020*), partly due to the lack of transgenic approaches allowing for direct functional testing of potential regulatory elements. Hence, other analytical methods must be used to discover enhancer regions de novo across the genome. A set of putative planarian TFs have been identified based on available transcriptomic data, but genome-wide TF interaction with potential enhancers has not been elucidated (*Swapna et al., 2018*; *Pascual-Carreras et al., 2020*). Ultimately, TF interactions with enhancers can be used to construct gene regulatory networks (GRNs) (*Duren et al., 2017*; *Lowe et al., 2019*; *Miraldi et al., 2019*, *Duren et al., 2020*; *Janssens et al., 2022*).

In planarians, regeneration is based on a population of somatic pluripotent stem cells called neoblasts that can differentiate into all cell types of the adult body plan and are the only cycling cells in the adult worm (*Aboobaker, 2011*; *Rink, 2013*, *Zhu and Pearson, 2016*). Nonetheless, how neoblasts dynamically retain pluripotency, specify fate, and differentiate remains unknown from a regulatory network perspective, while a number of different models at the cell biological level have been hypothesized (*Adler and Sánchez Alvarado, 2015*, *Swapna et al., 2018*; *Raz et al., 2021*). Fluorescence-activated cell sorting (FACS) separates cells into three compartments: S/G2/M cell-cycle stage neoblasts (known as X1), G1 stage neoblasts and postmitotic progeny (known as X2), and differentiated cells (known as Xins, as the insensitive to radiation treatment) (*Hayashi et al., 2006*; *Labbé et al., 2012*; *Onal et al., 2012*; *Romero et al., 2012*; *Solana et al., 2012*). Analysis of the X1 cells has been used to catalog genes expressed in neoblasts, and these studies have revealed that neoblasts are heterogeneous and can be subdivided into multiple classes based on their gene expression profiles (*van Wolfswinkel et al., 2014*; *Fincher et al., 2018*; *Plass et al., 2018*). Some classes of fate-specified neoblasts are termed specialized neoblasts, which include precursors to eyes, protonephridia, epidermis, intestine, pharynx, neurons, and muscle (*van Wolfswinkel et al., 2014*; *Raz et al., 2021*). Fate-specific transcription factors (FSTFs) are expressed in S/G2/M phase and are thought to direct fate specification into different cell types (*Raz et al., 2021*). Evidence from current experimental data supports a model where specialized neoblasts can divide asymmetrically giving rise to one fate-specified, postmitotic daughter and a proliferative neoblast that may still specify different fates (*Raz et al., 2021*). Genome-wide identification of active enhancers in the X1 compartment would shed light on the GRNs regulating the dynamic behavior of neoblasts in planarian regeneration (*Labbé et al., 2012*; *Onal et al., 2012*; *Solana et al., 2012*). Currently, no such predictions of enhancers or GRNs of TFs exist.

Identifying enhancers is challenging and especially so in non-model organisms, but various genome-wide high-throughput sequencing techniques have revealed signatures indicative of enhancers (*Shlyueva et al., 2014*, *Tomoyasu and Halfon, 2020*). Chromatin accessibility has proven to be a universal attribute of active enhancers and other regulatory regions in eukaryotes (*Thomas et al., 2011*; *West et al., 2014*; *Zhu et al., 2015*; *Daugherty et al., 2017*; *Klemm et al., 2019*). Assay for transposase-accessible chromatin using sequencing (ATAC-seq) has become the standard method of mapping open chromatin regions in various taxa (*Buenrostro et al., 2013*; *Li et al., 2019*; *Yan et al., 2020*). However, chromatin accessibility is not unique and specific to enhancers, and thus other complementary methods are used to discriminate enhancers from other open chromatin regions, particularly at the point of initial identification.

Enhancers are known to be flanked by nucleosomes with specific histone modifications (*Shlyueva et al., 2014*). In both mammals and *D. melanogaster*, acetylated lysine 27 of histone H3 (H3K27ac) is known to mark active enhancers together with mono-methylation of lysine 4 on histone H3 (H3K4me1), whereas H3K4me1 alone marks predetermined or poised enhancers (*Heintzman et al.,*

*2007*; *Creyghton et al., 2010*; *Ernst et al., 2011*; *Rada-Iglesias et al., 2011*; *Bonn et al., 2012*; *Arnold et al., 2013*; *Calo and Wysocka, 2013*). However, poised enhancers are not always activated, most enhancers are activated without a prior poised state, and some poised enhancers are later actively repressed, indicating that the poised enhancer state is not necessarily indicative of a pre-activation state (*Bonn et al., 2012*; *Rada-Iglesias et al., 2011*; *Koenecke et al., 2017*). Nonetheless, the bivalent epigenetic signature of H3K27ac and H3K4me1 seems to be a conserved indicator of active enhancers in metazoans and has been successfully used for enhancer detection in non-model organisms (*Gaiti et al., 2017a*, *Jänes et al., 2018*). In planarian neoblasts, the same histone modifications mark the active, suppressed, and bivalent state of promoters as in vertebrates (*Dattani et al., 2018*). Taken together, these data suggest that epigenetic marks may have a conserved association with regulatory elements across bilaterians (*Schwaiger et al., 2014*; *Sebé-Pedrós et al., 2016*; *Gaiti et al., 2017b*, *Dattani et al., 2018*).

Here, we significantly improve upon the annotation of the planarian genome (*Grohme et al., 2018*), take a computational approach to identify all TFs in this annotation, identify putative enhancers in the planarian genome supported by multiple lines of evidence, and then construct hypothetical GRNs active in neoblasts. We find that multiple enhancers have evidence of FSTF-mediated regulation, supporting the view that fate specification occurs in the S/G2/M phases. The FSTFs of different cell types seem to cross-regulate each other, revealing the potential for a dynamic GRN governing neoblast fate specification. We identified enhancers linked to several unstudied TFs implicating them in potentially central roles in neoblast TF GRNs. Enhancers linked to well-known planarian positional genes suggest regulatory mechanisms for some of the known links between these genes implicated by phenotypic studies. Finally, we show that our GRNs predict TF regulatory interactions implicated by available data measuring gene expression changes after TF RNAi. Overall, this work provides a foundation for future work on the regulatory logic of planarian stem cell biology and identifies many candidate TFs with predicted roles in regulating adult stem cells.

## Results

### Annotation with a full range of transcriptome samples identifies more than 3000 new protein coding genes in the *Schmidtea mediterranea* genome

We refined and extended the current annotation of the *S. mediterranea* genome (SMESG.1 genome and SMESG high-confidence annotation at Planmine; *Brandl et al., 2016*). We performed a genome-directed annotation based on the genome sequence and 183 independent RNA-seq datasets, including data both from whole worms and cell compartments (*Figure 1A and B*; this new annotation has also been used in *García-Castro et al., 2021*, but is described in more detail here). By including a diverse and large set of RNA-seq data, we sought to characterize transcripts undetected in individual studies and annotation attempts (*Hoff and Stanke, 2013*; *Hoff and Stanke, 2019*). Furthermore, we calculated proportional expression values for each cell population defined by FACS using approaches established previously (*Figure 1A*; *Dattani et al., 2018*).

In total, our expression-driven annotation process identified 91,068 transcripts at 28,097 genomic loci (*Figure 1B*). The annotation process validated all gene models in the SMESG high-confidence annotation, as all loci, transcripts, and exons were also found in our new annotation. In total, 50,213 transcripts were identified as putative new isoforms of previously identified loci (*Figure 1C*). Furthermore, 7412 new loci with 10,636 transcripts were found (*Figure 1B, C*). The protein coding potential of these new transcripts was assessed by defining putative open-reading frames (ORFs) and scanning for protein structures (*Figure 1B*). In total, 3121 new loci with 4752 transcripts were predicted to be coding, while 4291 loci with 5884 transcripts were predicted to be non-coding (*Figure 1B, C*).

The newly described isoforms across the genome were slightly shorter than known transcripts (median length 1618 bp vs. 1656 bp), while the new coding and non-coding transcripts were much shorter (median length 583 and 388 bp, respectively) (*Figure 1D*). However, the mean transcripts per kilobase million (TPM) value measured across the RNA-seq samples for new isoforms did not differ much from the levels of previously known transcripts, while the mean TPM values for new coding and non-coding transcripts were slightly higher (*Figure 1—figure supplement 1A*). This suggests that the main advantage of our approach was to discover shorter transcripts (and encoded proteins) not found

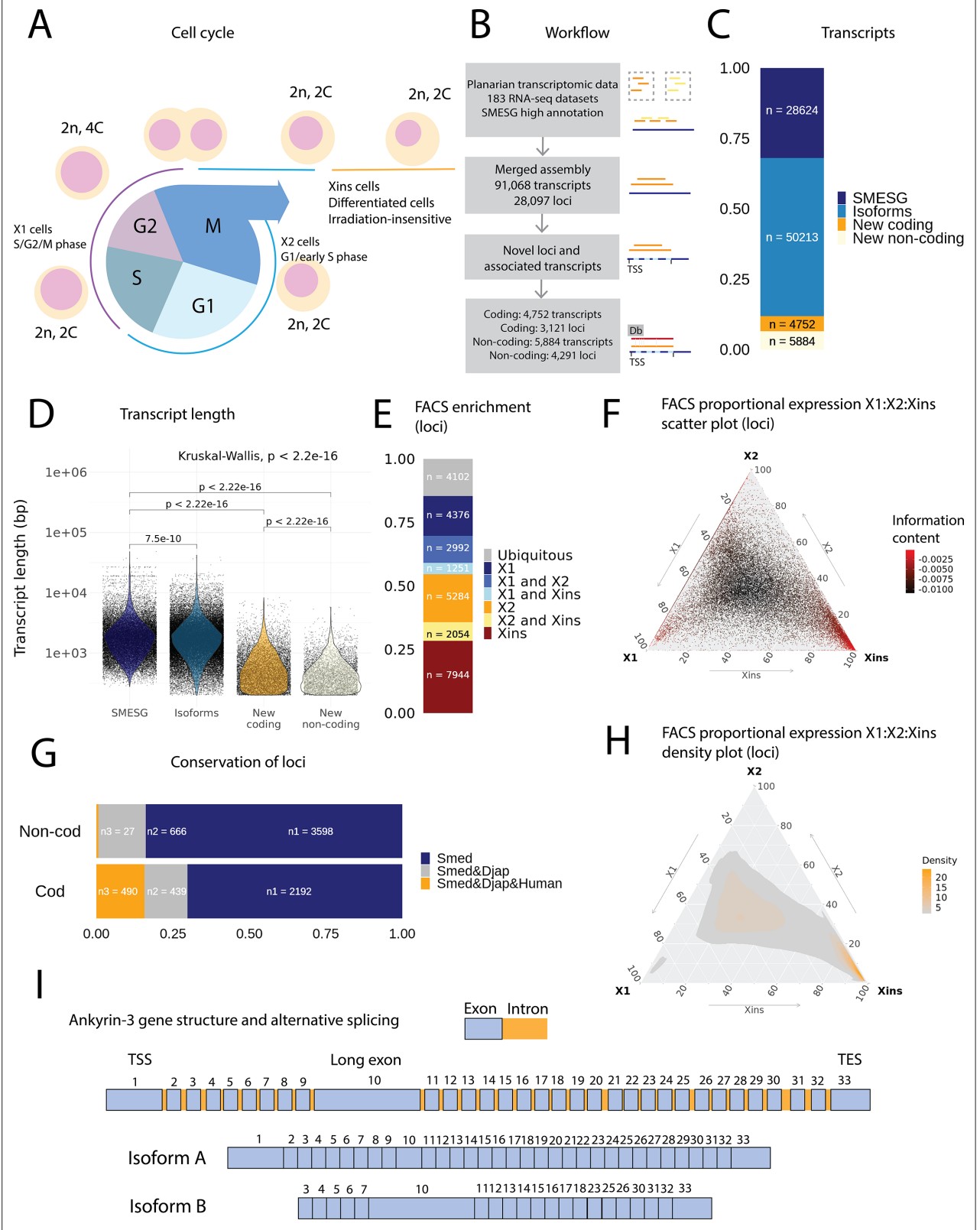

**Figure 1.** An expression-driven genome annotation of *Schmidtea mediterranea*. (**A**) Diagram of the planarian neoblast cell cycle illustrating how the fluorescence-activated cell sorting (FACS) cell compartments X1, X2, and Xins relate to different cell-cycle phases. (**B**) Overview of methodology for annotating the *S. mediterranea* genome and identifying novel coding and non-coding transcripts. In total, 183 RNA-seq datasets were aligned to the SMESG.1 genome (Planmine). Novel transcripts were assembled by using the SMESG-high-confidence annotation as a reference (Planmine).

*Figure 1 continued on next page*

*Figure 1 continued*

(**C**) The proportion of transcript identities in the new annotation. 'SMESG' are transcripts present in the known SMESG high-confidence annotation, 'Isoforms' are transcripts that are isoforms or splicing variants of known transcripts in the SMESG high-confidence annotation, and 'New coding' and 'New non-coding' are previously unknown transcripts deemed to have coding and non-coding potential, respectively. (**D**) The transcript lengths of known transcripts (SMESG, n = 28624), isoforms (n = 50213), new coding transcripts (n = 4752), and new non-coding transcripts (n=5884). Kruskal-Wallis test is used to compare all groups and Wilcoxon rank-sum test is used for pairwise comparisons. (**E**) FACS categorization of 28,003 total loci into enrichment groups (***Dattani et al., 2018***). The enrichment groups are X1 (X1 proportional expression ≥ 50%), X2 (X2 proportional expression ≥ 50%), Xins (Xins proportional expression ≥ 50%), X1 and X2 (X1 + X2 proportional expression ≥ 75%, neither enriched in X1 nor X2), X1 and Xins (X1 + Xins proportional expression ≥ 75%, neither enriched in X1 nor Xins), X2 and Xins (X2 + Xins proportional expression ≥ 75%, neither enriched in X2 nor Xins), and ubiquitous (loci not categorized into the enrichment groups above and with roughly equal proportion in all three groups). (**F**) Ternary plot of proportional expression values in the X1, X2, and Xins cell compartments. Each dot represents one single locus. The information content metric represents the enrichment of a locus to one of the three compartments, meaning that gray loci are unspecific while red loci are specific to a single compartment (see 'Materials and methods' for details). (**G**) Conservation of new coding and non-coding loci to loci in *Dugesia japonica* and humans. 'Smed' (n1) are loci exclusive to *S. mediterranea*, 'Smed&Djap' (n2) are loci with a homolog in *D. japonica* but not in humans, and 'Smed&Djap&Human' (n3) are loci with homologs both in *D. japonica* and humans. (**H**) Ternary plot of the density distribution of proportional expression values in the X1, X2, and Xins cell compartments. The plot gives a visual condensation of ternary plot F. The density distribution is estimated by two-dimensional kernel density estimation. (**I**) Schematic of the gene structure of ankyrin-3 and two example isoforms that are alternatively spliced (SMEST010564001.1.p1 and MSTRG.3975.46.p1). The detailed genomic track of all the isoforms is given in ***Figure 1—figure supplement 1F***. TSS, transcriptional start site; TES, transcript end site.

The online version of this article includes the following figure supplement(s) for figure 1:

**Figure supplement 1.** Information on the genome annotation of *Schmidtea mediterranea*.

by previous annotation approaches (***Figure 1D***, ***Figure 1—figure supplement 1***). We sorted all annotated genes to FACS enrichment groups (***Figure 1H***), using previously described methods (***Dattani et al., 2018***). The number of genes with enriched expression was highest in the Xins compartment and lowest in X1 cells (***Figure 1E***). As expected by overall lineage relationships, the shared enrichment between X1 and Xins is less common than between X1 and X2 and between X2 and Xins cells, congruent with the fact that X1 cells are all neoblasts, X2 is an amalgam of G1 neoblasts and differentiating postmitotic progeny, and Xins cells are their collective differentiation products (***Figure 1E***). By inspecting the distribution of proportional expression values, we also see that the distribution is shifted towards the Xins compartment (***Figure 1F, H***), thus overall expression of genes is enriched in differentiated cell types.

The majority of the new coding loci were unique to *S. mediterranea* (70 %), but 30% had a homolog in the *Dugesia japonica* genome (E-value cutoff $10^{-10}$, ***Figure 1G***; ***An et al., 2018***). Only 27 non-coding loci (0.6 %) had a potential homolog in the human genome, but 666 loci (16 %) had a homolog in the *D. japonica* genome (***Figure 1G***). The Gene Ontology (GO) analysis of new coding loci revealed that the new genes were enriched for a function in RNA biosynthesis, metabolic processes, dephosphorylation, deacetylation, transmembrane transport, and mitochondrial functions (***Figure 1—figure supplement 1C–E***). These data suggest that many of our newly annotated genes are conserved amongst the group of planarians used for regeneration and stem cell research.

**Table 1.** Genes with more than 100 transcript isoforms.

| Number of isoforms | Gene | Description |
| --- | --- | --- |
| 571 | TMEM25 | Transmembrane protein 25 |
| 291 | CLEC18B | C-type lectin domain family 18 member B |
| 206 | ANK2 | Ankyrin 2 |
| 143 | GABPB2 | GA binding protein transcription factor subunit |
| 138 | PACRG | Parkin coregulated |
| 112 | ANK3 | Ankyrin 3 |
| 106 | DST | Dystonin |
| 104 | ANK3 | Ankyrin 3 |

Some genes in the new annotation displayed high levels of alternative splicing, including homologs of lectin, ankyrins, and dystonin (*Figure 1I*, *Figure 1—figure supplement 1B*, *Table 1*). In mammals, ankyrin-3 (ankyrin-G) is a structural protein localized to the axon initial segment (AIS) and the nodes of Ranvier, and alternative splicing is known to underlie its functional diversity (*Figure 1I*, *Figure 1—figure supplement 1F*; *Hopitzan et al., 2005*; *Lopez et al., 2017*; *Nelson and Jenkins, 2017*). A planarian homolog of ankyrin-3 has 112 isoforms, including isoforms with one long exon, supporting the finding that giant ankyrin-based cytoskeleton of the AIS may have been present in the last common ancestor of bilaterians (*Figure 1I*, *Figure 1—figure supplement 1F*; *Jegla et al., 2016*).

Taken together, these initial example analyses of our new annotation, particularly the discovery of many hundreds of new loci and thousands of putative alternative isoforms, suggest that it will have an important utility for the research community studying all aspects of planarian biology.

## Comprehensive annotation of planarian transcription factors highlights a diversity of unknown zinc fingers

We screened for TFs in the new annotation using the same approach as *Swapna et al., 2018* (*Figure 2A*). We validated the TF potential through a systematic protein domain annotation, assessed for homology to known TFs, and manually reviewed the list of TFs to assign them to planarian TFs present in the literature and databases (*Figure 2A*). Altogether, we found predicted 551 TFs in *S. mediterranea*, of which we found 248 to be described in the planarian literature as a named molecular sequence (i.e., the sequence was assigned a proper TF name; for details, see 'Materials and methods'). The naming of planarian TFs in the literature was mostly consistent, but some inconsistencies were found (*Supplementary file 1*). We classified the TFs into four structural categories: basic domains, zinc domains, helix-turn-helix domains, and other domains (*Stegmaier et al., 2004*). Most basic domains had been described by *Cowles et al., 2013*, but we still identified new homologs of *Atf*, *Batf*, *Creb*, *Htf*, *Mad*, *Matf*, *MyoD*, *Npas*, and *Pdp* family members, each with broad established roles in metazoan biology (*Figure 2B*, *Supplementary file 1*).

In contrast, we identified multiple uncharacterized zinc finger domain TFs (ZNFs), many of which have not received much consideration yet in planarian regeneration research. While some of these unstudied ZNFs could be assigned to well-known ZNF families such as GATA, KLF, EGR, and PRDM, many could not be assigned to well-described families. The nomenclature of ZNFs was based on the naming of human proteins to which they have the highest identity, and hence many appear in the ZNF and ZNP categories (*Figure 2C*). Interestingly, we also find ZNFs related to SCAN-domain containing zinc fingers (ZSCAN) and pogo transposable elements with KRAB domain (POGK) previously only described in vertebrates (*Emerson and Thomas, 2011*; *Gao and Qian, 2020a*). However, these planarian ZNFs do not contain the SCAN or KRAB domains, and the similarity arises from the DNA-binding domains. While members of the pogo transposable element superfamily are found throughout the metazoans, the KRAB subfamily is specific to vertebrates (*Gao et al., 2020b*). For this reason, we have provisionally named these ZNFs Zscan-like and POGK-like (*Figure 2C*).

For helix-turn-helix domains, new homologs were found for several described TF families, including *Nkx* and *Six*, and a few TFs belonging to families not previously described in planarians were also discovered, such as *Lmx* and *Shox* (*Figure 2D*). Some TFs were also newly annotated for other domain families (*Figure 2E*).

We proceeded to allocate proportional expression values to the TFs with respect to the X1, X2, and Xins cell compartments (*Dattani et al., 2018*). The distribution of the proportional expression values was even, although a slight skew towards the X1 compartment was evident (*Figure 2F, G*). Most TFs were assigned to the X1 and Xins compartments and the least to the X2 compartment (*Figure 2F, G*). In other words, few TFs were specific to the X2 compartment, while most TFs were specifically expressed in stem cells and their nascent progeny (X1 and X2) or in progeny and differentiated cells (X2 and Xins). This would be expected by overall lineage relationships. There was no compartment-specific enrichment for different TF domains ($\chi^2$ test, p=0.2).

We then moved to assigning predicted target binding motifs to the annotated set of planarian TFs. Most studies in non-model organisms have tended to use motifs directly from only a single-model organism or solely rely on de novo motif enrichment without reference to the TFs actually present in the studied organism (*Gaiti et al., 2017a*, *Gehrke et al., 2019*; *Murad et al., 2019*). However, we used the same approach as *Siebert et al., 2019* and searched the JASPAR database for TFs with the

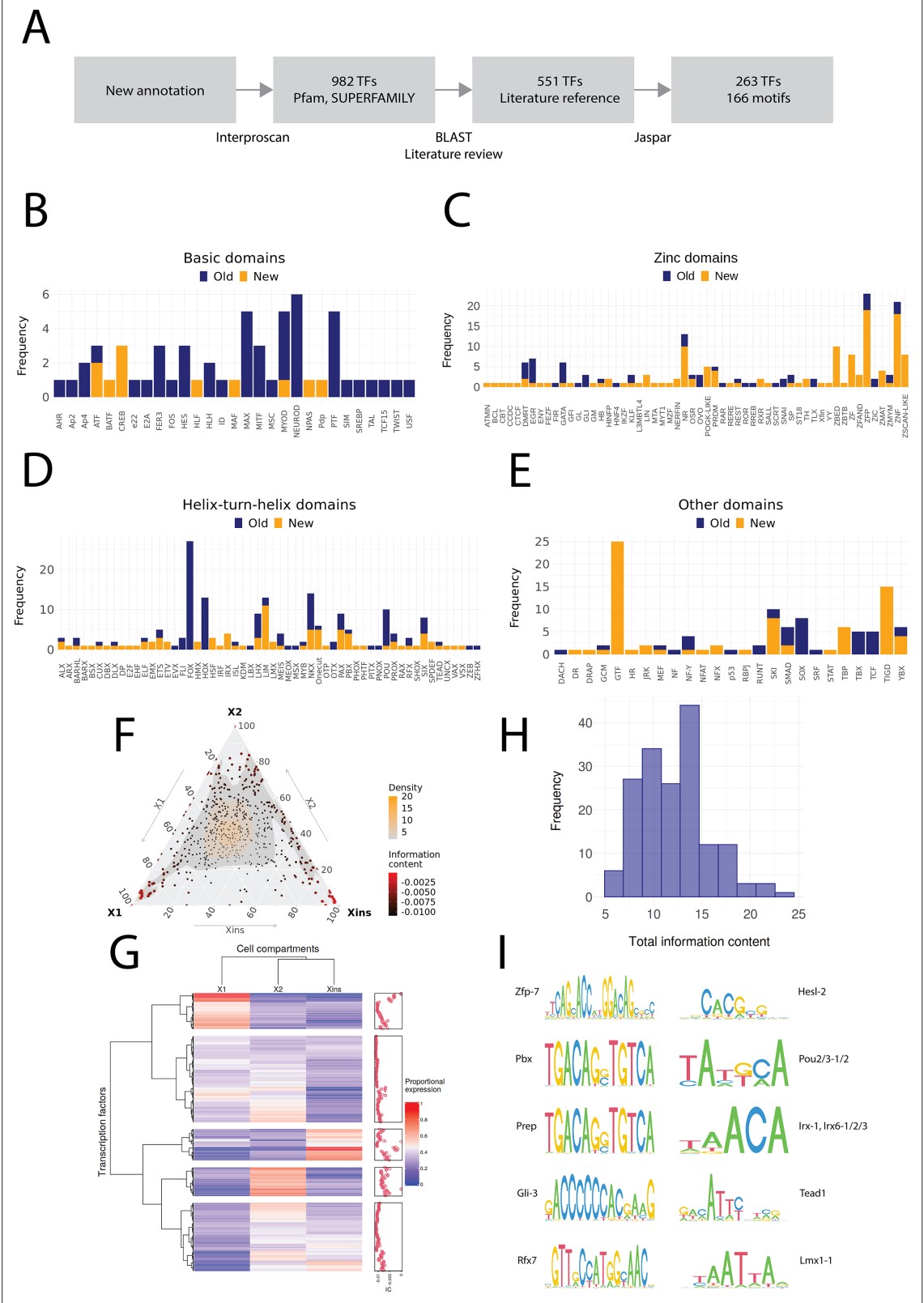

**Figure 2.** Annotation of planarian transcription factors. (**A**) Diagram of the transcription factor (TF) annotation process. The coding transcripts identified in the new genome annotation were screened for conserved TF structures by using Interproscan and the Pfam and SUPERFAMILY databases. Homologs were then searched for the TFs with BLAST, and the planarian literature was reviewed to assign screened TFs to previously described. The manually reviewed TF proteins were then used to predict motifs. (**B–E**) Identified TF families with basic domains (**B**), (**C–E**). 'Old' refers to TFs present in the

*Figure 2 continued on next page*

*Figure 2 continued*

literature, while 'New' are TFs without a reference in the literature or new additional homologs. (**F**) Ternary plot of proportional expression values in the X1, X2, and Xins cell compartments. Each dot represents one single TF. The information content metric represents the enrichment of a locus to one of the three compartments (see 'Materials and methods' for details). (**G**) Heatmap and hierarchical clustering of proportional fluorescence-activated cell sorting (FACS) expression profiles for all TFs. (**H**) Histogram of the total information content of the identified motifs. (**I**) Example sequence logos for some of the best characterized motifs. The motifs of pbx (MA0782.1) and prep (MA0783.1) have almost identical PWMs, but they are listed as separate motif entries in JASPAR.

highest similarity to predict motifs for planarian TFs (*Figure 2A*). In total, we found 166 motifs that were assigned to 263 TFs with a normal distribution of motif information value (*Figure 2H*, *Supplementary file 1*). The most informative motifs were found for *zfp-7*, *pbx*, *prep*, *gli-3*, and *rfx7*, while the least informative motifs were for *hesl-2*, *pou2/3*, *irx*, *tead1*, and *lmx1* (*Figure 2I*).

## Histone modifications and chromosome accessibility mark enhancer-like regions

To identify potential enhancer regions, we analyzed previously generated ChIP-seq data with respect to the enhancer-associated histone modification H3K4me1 in X1 cells (*Mihaylova et al., 2018*). Furthermore, we sequenced the epigenome of X1 cells with respect to the histone modification H3K27ac to identify genomic regions indicative of an active enhancer state. H3K27ac was enriched at promoter regions, suggesting that this epigenomic feature for this histone modification is conserved in planarians (*Figure 3—figure supplement 1A, B*; *Gaiti et al., 2017b*). We identified 37,345 H3K27ac peaks and 13,868 H3K4me1 peaks that were generally less than 200 bp wide (*Figure 3A, B*). At H3K4me1 peaks, the H3K27ac signal was strongest at the peak center at almost all peaks, while at H3K27ac peaks the H3K4me1 displayed a bimodal peak at the peak center of most peaks (*Figure 3C, D*). This pattern of H3K4me1 flanking H3K27ac peaks at enhancers has been previously described in mammals, and our data suggest that this may be evolutionarily conserved within metazoans (*Gorkin et al., 2012*; *Spicuglia and Vanhille, 2012*; *Pundhir et al., 2016*).

As our ChIP-seq data followed well-established enhancer-like patterns, we used the ChIP peaks to select putative enhancer-like regions. We calculated a mean peak value at all peaks with respect to the H3K27ac and H3K4me1 signal (*Figure 3E*) and selected all H3K27ac peaks that were at most 500 bp from H3K4me1 peaks and determined these 5529 peaks to be an initial set of putative active enhancer-like regions in cycling adult stem cells. The H3K27ac and H3K4me1 signals correlated at these enhancer-like regions more than at the other peaks in the genome (*Figure 3F*).

Furthermore, we calculated the mean peak value with respect to the change in H3K4me1 signal upon RNAi-mediated *Smed-lpt* knockdown (corresponding to the N-terminus of mammalian mll3/4 or kmt2c/d, see *Mihaylova et al., 2018*). In mammals, Mll3 and Mll4 are two paralogous methyltransferases of the COMPASS family (SET1/MLL) that regulate enhancer activity by mono-methylating H3K4 (*Wang et al., 2021*). In addition, the mll3/4 methyltransferase complex associates with the histone acetyltransferase p300/CBP that mediates H3K27 acetylation at enhancers and thus gives rise to the active enhancer landscape (*Wang et al., 2021*). The enhancer-like regions were clearly more responsive to the knockdown of *Smed-lpt* as compared to random points within the genome, and the response was most evident at the center of putative enhancer-like regions (*Figure 3G, H and K*). Thus, the H3K4me1 reduction after *Smed-lpt* knockdown provides further functional support for the identification of active enhancers in planarian stem cells as these are targets for active histone methylase activity associated with enhancers.

We optimized and performed ATAC-seq on X1 cells to measure high-resolution chromatin accessibility in conjunction with the histone modifications (*Buenrostro et al., 2013*) as a potential further source of evidence. Enhancer-like regions identified by ChIP-seq analysis had a more accessible chromatin configuration than random points in the genome, and the peaks of open regions were positioned at the center of the predicted enhancer-like regions implicated by ChIP-seq data (*Figure 3I, J and L*). Thus, ATAC-seq provides independent evidence for these regions being putative active enhancers in planarian stem cells.

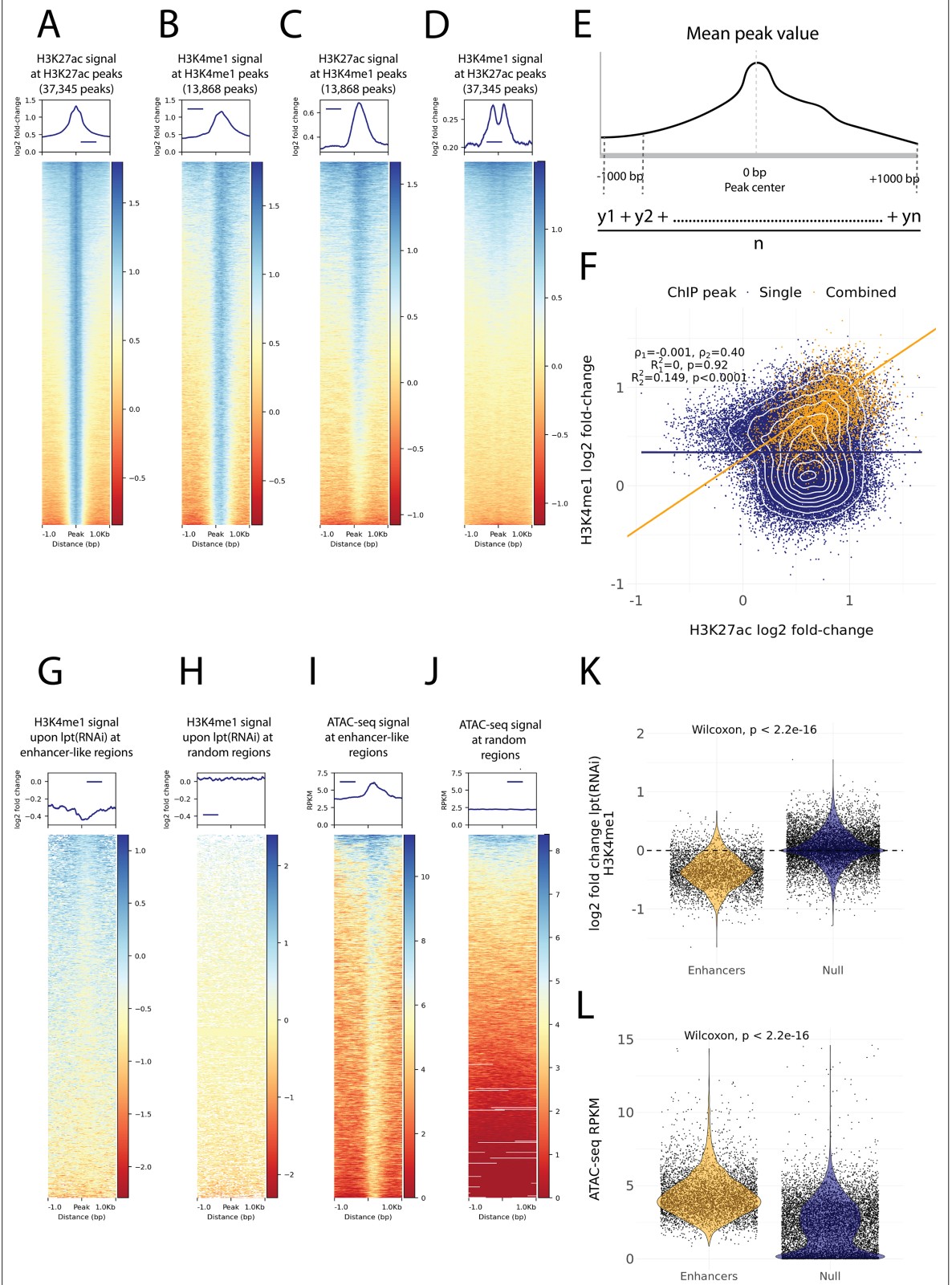

**Figure 3.** Histone modifications and chromatin accessibility of enhancer-like regions. (**A**) H3K27ac ChIP-seq signal around independently called H3K27ac peaks. (**B**) H3K4me1 ChIP-seq signal around independently called H3K4me1 peaks. (**C**) H3K27ac signal around the same H3K4me1 peaks as in (**B**). The signal centers around the center region. (**D**) The H3K4me1 signal around the same H3K27ac peaks as in (**A**). The signal displays a bimodal shape around the center region. The signal of the y-axis and the heatmap in (**A**–**D**) is given as the log2 fold-change relative to the input and the peak width is

*Figure 3 continued on next page*

*Figure 3 continued*

2000 bp. (**E**) A visual diagram of the calculation of the mean peak value. Mean peak value is defined as the mean of signal values taken at an interval of 10 bp from 1000 bp upstream to 1000 bp downstream of the peak center. (**F**) Scatter plot of H3K27ac and H3K4me1 log2 fold-change mean peak values at the ChIP-seq peaks. H3K4me1 peaks that are at most 500 bp from a H3K27ac peak are defined as combined peaks and putative enhancer-like regions (model 2, orange), while the non-overlapping H3K4me1 and H3K27ac peaks are referred to as single peaks (model 1, navyblue). (**G**) The change in the H3K4me1 signal upon *lpt* (mll3/4) RNAi at the combined peaks or putative enhancer-like regions. The change is strongest at the peak center. The log2 fold-change is calculated as the RNAi value relative to the wildtype value. (**H**) The H3K4me1 *lpt(RNAi)* signal at 10,000 random regions in the genome. (**I**) The ATAC-seq signal at the combined peaks or putative enhancer-like regions. The signal is given as reads per kilo base per million mapped reads (RPKM) (**J**) The ATAC-seq signal at 10,000 random regions in the genome. (**K**) Violin plot of H3K4me1 *lpt(RNAi)* signal at putative enhancer-like regions (n = 5529) and random regions (n = 10,000) in the genome. The H3K4me1 signal is reduced at enhancers. Wilcoxon rank-sum test is used for pairwise comparisons. (**L**) Violin plot of the ATAC-seq signal at putative enhancer-like regions (n = 5529) and random regions (n = 10,000) in the genome. The ATAC-seq signal is higher at enhancers. Wilcoxon rank-sum test is used for pairwise comparisons.

The online version of this article includes the following figure supplement(s) for figure 3:

**Figure supplement 1.** Histone modifications at transcription start sites and footprinting scores in relation to enhancer-like regions.

## Gene regulatory networks involving fate-specific transcription factors in neoblasts

Having defined enhancer-like regions (*Supplementary file 2*), integration with both TF and potential target gene expression data allowed us to begin constructing preliminary GRNs to demonstrate the utility of our neoblast enhancer predictions. First, we assigned enhancer-like regions to their closest gene, assuming a high probability that these will be the putative target genes of enhancer-like regions (*Supplementary file 2*). The distance from the enhancer-like regions to the target gene transcription start site (TSS) varied from being proximal to the promoter region to being as far as 89 kb away (*Figure 4A*). We found that TFs themselves were significantly enriched within the set of all predicted target genes (*Figure 4B*). Furthermore, these TFs were in turn enriched for the 43 FSTFs previously shown to be expressed in S/G2/M neoblasts (*Figure 4B*; *Raz et al., 2021*), again suggesting that our predicted set of enhancers are real. In addition to TF activity, the predicted target genes linked to enhancer-like regions were enriched for RNA metabolic and RNA biosynthetic processes and other biosynthetic processes, as well as transcription and regulation of gene expression (*Figure 4C*). These data suggest that our enhancer predictions include real enhancers involved in regulating key aspects of neoblast biology.

To establish more persuasive and direct regulatory links between TFs and enhancer-like regions, we used ATAC-seq footprinting to detect potentially bound motifs in the planarian stem cell genome (*Bentsen et al., 2020*; *Figure 4D*, *Supplementary file 3*). ATAC-seq footprints are short inaccessible or less accessible regions within an otherwise accessible region, indicative of DNA binding by a TF or another DNA-binding protein (*Bentsen et al., 2020*; *Figure 4D*). As for the raw ATAC-seq signal, footprint scores were higher in enhancer-like regions than in random regions of the genome (*Figure 3—figure supplement 1C, D*). Overall footprinting analysis found 22,489 putatively bound TF motifs in the enhancer-like regions but no bound motifs in random regions, providing further support that these regions are potential enhancers. The TFs with the most binding sites predicted by footprint analysis in the genome overall were *irx-1/irx6-1/2/3*, *egr4-1/2*, *smad1/9*, *sp5*, and *nf-ya1/2*, while the TFs with the least bound motifs were *mef2-1*, *mef2-2*, *phox*, *pou6-2*, and *lmx1-2* (*Figure 4E* *Supplementary file 3* and *Supplementary file 4*). Overall, these data provide a resource of studying GRNs and specific regulatory interactions in planarians neoblasts. We found no obvious difference in the distance of predicted TF footprints from TSSs for different TFs (*Figure 4—figure supplement 1*).

To test the utility of these data for building and testing GRNs, we focused on the previously defined FSTFs, established the regulatory links between them, and constructed a putative stem cell GRN based on our datasets (*Figure 4F*, *Supplementary file 5*). In total, we could include 35/43 FSTFs into a GRN prediction and found evidence for multiple cross-regulatory links that may serve to allow stem cells to decide between fates. for example, neural FSTFs regulate enhancers of muscle FSTFs and vice versa (*Figure 4F*). The interactions in this GRN allow specific hypotheses to be formed and tested in the future. As examples, we present in detail the genomic region of the epidermal FSTF *p53*, the muscle FSTF *myoD*, and the neural FSTFs *nkx6-1* and *soxB1-1* (*Figure 5A–D*). The epidermal FSTF *p53* has multiple TFs predicted to bind at the promoter region, including a muscle FSTF (*myoD*) (*Scimone et al., 2014*; *Scimone et al., 2017*), and TFs relating to position (*zic-1*) (*Vásquez-Doorman*

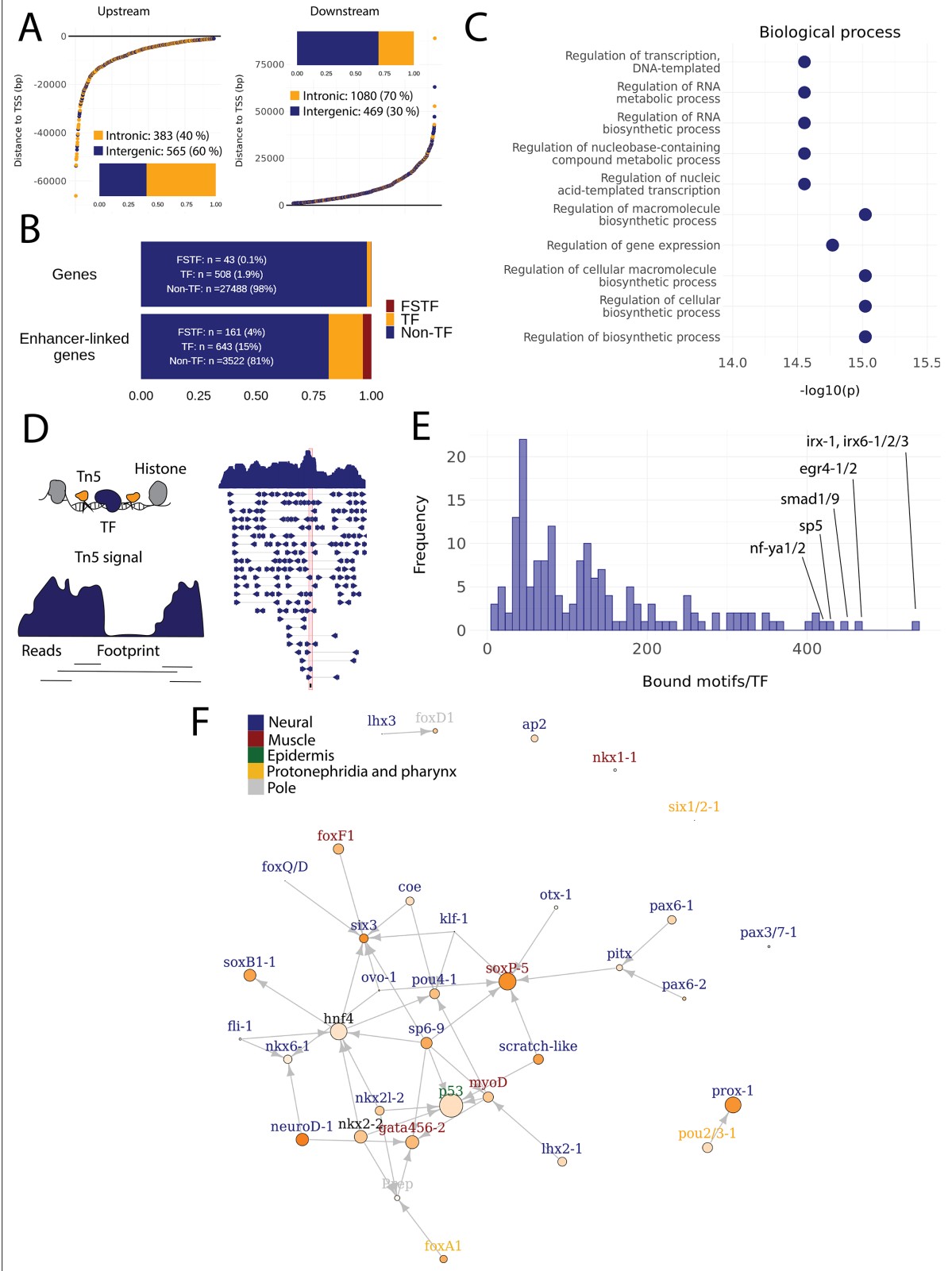

**Figure 4.** Predicted gene regulatory networks involving fate-specific transcription factors in neoblasts. (**A**) Distance of enhancer-like regions to the transcription start site (TSS) of nearest genes. The nearest genes are assumed to be the putative target genes of the enhancer-like regions. (**B**) The proportion of fate-specific transcription factors (FSTFs), transcription factors (TFs), and non-TF genes of all planarian genes and of the target genes linked to enhancer-like regions. FSTFs and TFs are enriched in the set of target genes ($\chi^2$ = 2660, p<0.001). (**C**) Gene Ontology (GO) analysis of

*Figure 4 continued on next page*

*Figure 4 continued*

biological processes of enhancer-linked target genes compared to all genes in the planarian genome. (**D**) Schematic overview of ATAC-seq footprinting (***Bentsen et al., 2020***). Both the tagmenting enzyme Tn5 and TF bind to accessible regions of the genome. Footprints are short and sharp inaccessible segments in otherwise accessible regions. The genomic track figure represents an example of ATAC-seq reads and footprint score overlapping with an *sp5* motif in the genome. (**E**) Histogram of the total number of bound motifs in the genome for each TF. The TFs with most bound motifs in the genome are marked. (**F**) Putative gene regulatory network (GRN) of FSTFs in neoblasts. The epidermal FSTFs are labeled in green, the pole-associated FSTFs are labeled in gray, the muscle FSTFs are labeled in red, the intestinal FSTFs are labeled in black, and the neural FSTFs are labeled in blue. The size of the nodes reflects the absolute expression in X1 cells (transcripts per million reads [TPM]), and the color of the node reflects the proportional expression in X1 cells (the more orange, the higher proportional expression). The arrows represent a regulatory link but do not discriminate between positive and negative interactions.

The online version of this article includes the following figure supplement(s) for figure 4:

**Figure supplement 1.** Distance of bound transcription factor footprints to transcriptional start sites.

*and Petersen, 2014*) and cell migration (*zeb-1* and *snail-1/2*) (***Abnave et al., 2017***; ***Figure 5A***). An enhancer-like region within an intron of the upstream gene *ythdc23* was predicted to be associated with the downstream muscle FSTF *myoD* as the TSS of *myoD* is closer of the two and *ythdc23* is lowly expressed in X1 and X2 cells. The enhancer-like region includes bound TF motifs such as the neoblast-enriched zinc finger *fir-1*, the muscle segment homeobox gene *msx*, the neural *pou-p1*, *sp6-9*, *lhx2-1*, and position-related *foxA(P)* (***Figure 5B***). A distal enhancer-like region was found downstream of *nkx6-1*, including potentially bound TF motifs such as the neoblast-enriched *egr-1*, the neural *ovo-1* and *fli-1*, the muscle-related *twist*, the position-related *smad4-1*, and the pigmentation-related *ets-1* (***Figure 5C***). A distal enhancer-like region was found upstream of *soxB1-1*, including bound TF motifs such as the position-related *smad1* and *foxA(P)* (***Molina et al., 2007***; ***Reddien et al., 2007***; ***Pascual-Carreras et al., 2020***), neural *da* (***Cowles et al., 2013***), intestinal *hnf4* (***Wagner et al., 2011***, ***Scimone et al., 2014***; ***van Wolfswinkel et al., 2014***), and *zeb-1* and *snail-1/2* relating to cell migration (***Abnave et al., 2017***; ***Figure 5D***). In the future, these putative regulatory links can be verified and studied by experiments using functional genomics involving the RNAi knockdown of individual genes and subsequent RNA-seq and ATAC-seq analysis.

## Multiple enhancers are linked to unstudied transcription factors

In addition to the relatively well-known planarian FSTFs (***Raz et al., 2021***), we investigated GRNs relating to less-well-studied TFs with multiple bound motifs in associated enhancer-like regions as these may also have a potentially central role in the neoblast regulation. We defined the number of enhancer-like regions, number of putative bound motifs, and number of putative unbound motifs for each TF (***Figure 6A***, ***Supplementary file 6***). In addition, we selected TFs expressed in the X1 compartment (proportional expression of X1 > 1/3) and constructed the putative GRN summarizing all the 502 regulatory links of these 109 TFs (***Figure 6B***). In the X1 compartment (X1% > 1/3), the least-well-studied TFs with the most bound motifs in enhancer-like regions were *znf596*, *tbx-20*, *hesl-1*, *atoh8-1*, and *ikzf1* (***Figure 6A, B***, ***Supplementary file 2*** and ***Supplementary file 6***). Outside the X1 compartment, *msx*, *zf-6*, *vsx*, and *pdp-1* had the most bound motifs (***Figure 6A***).

The TF with most regulatory interactions, *znf596*, has been characterized to be expressed in neoblasts and more specifically in the neoblasts committed to the neural fate, but otherwise its function is unknown (***Figure 6A–C***, ***Supplementary file 6***; ***Fincher et al., 2018***). We found that numerous FSTFs were predicted to bind to putative intronic enhancers within *znf596*, including neural FSTFs *sp6-9*, *neuroD-1*, and *ovo-1*, the intestinal FSTF *gata456-1* and/or the muscle FSTF *gata456-1*, and the position control genes (PCGs) *smad1/9* (BMP signaling), *isl-1*, and *prep* (***Figure 6C***). In single-cell transcriptomics data, the gene is clearly expressed in a subset of neoblasts, calling for further mechanistic studies of its regulatory function. The binding motif of *znf596* could not be predicted, so its target genes cannot be implicated within the current datasets and approaches.

We identified the same five planarian *tbx* genes that have been previously reported, namely, *tbx-1/10*, *tbx-2/3a*, *tbx-2/3b*, *tbx-2/3c*, and *tbx-20* (***Tewari et al., 2019***). In mammals, the *tbx* genes have multiple roles in development, and *tbx3* is involved in regulating pluripotency of embryonic stem cells (***Baldini et al., 2017***; ***Khan et al., 2020***). Nonetheless, the function of planarian *tbx* genes has not been clarified. Here, we found several predicted bound motifs of *tbx* genes in the genome (*tbx-1/10* 91 motifs, *tbx-2/3a/c* 81 motifs, *tbx-20* 62 motifs) and multiple bound motifs at enhancer-like regions

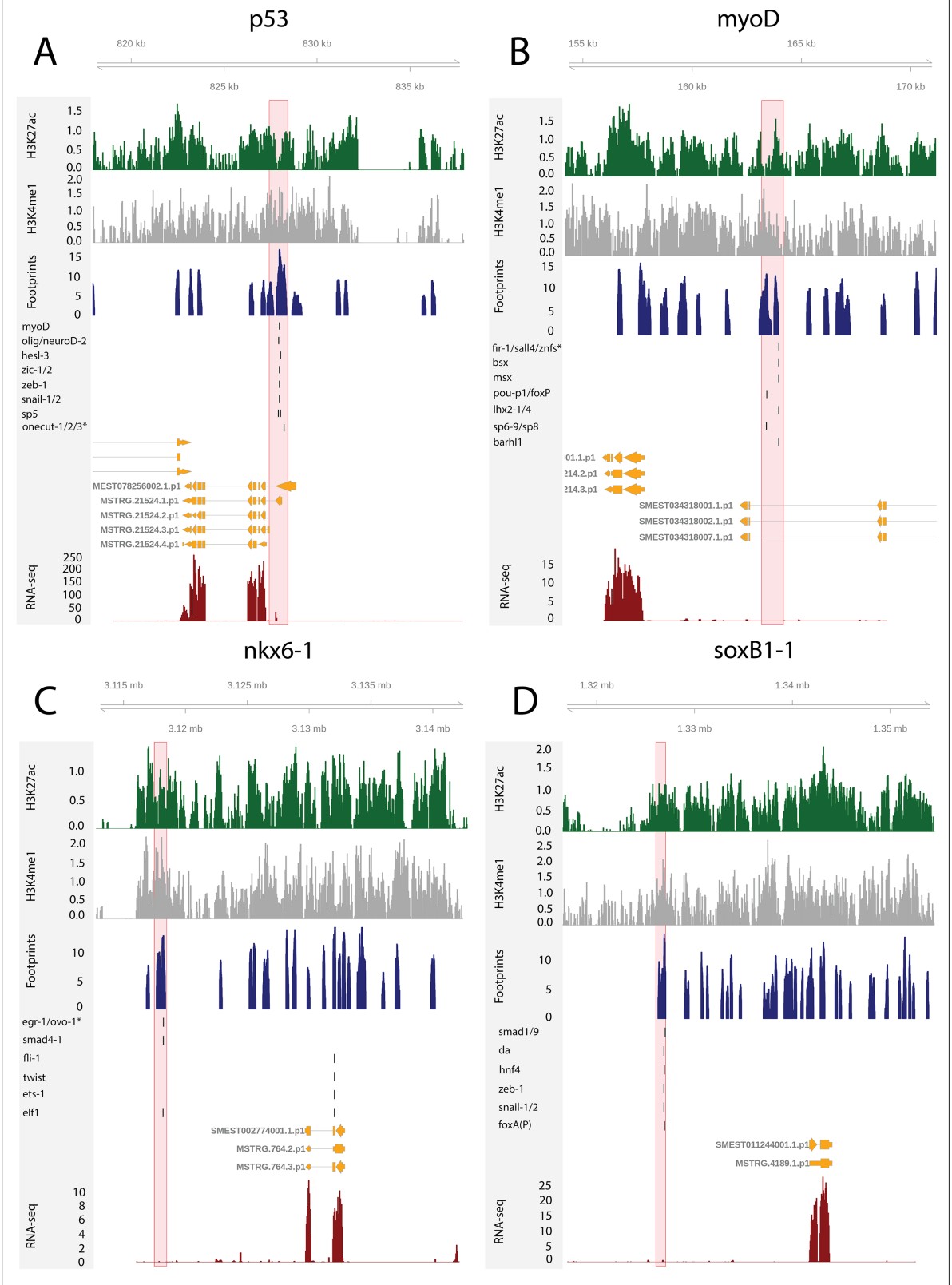

**Figure 5.** The genomic tracks of four fate-specific transcription factors (FSTFs) (**A–D**). The first track is the H3K27ac signal (log2 scale compared to the input sample), the second track is H3K4me1 signal (log2 scale compared to the input sample), the third track is the ATAC-seq footprint score (TOBIAS footprint score), the fourth track represents bound motifs in an enhancer-like region (as predicted by TOBIAS), the fifth track represents the transcripts of the gene in the new annotation, and the sixth track represents the gene expression level of the gene in X1 cells (RNA-seq fragments per kilobase

*Figure 5 continued on next page*

*Figure 5 continued*

of transcript per million [FPKM] mapped reads). (**A**) Genomic track of the epidermal FSTF *p53*. The promoter region marked in red (width 1000 bp) has multiple bound motifs. (**B**) Genomic track of the muscle FSTF *myoD*. An enhancer-like region marked in red (width 1000 bp) is located in the intron of the gene YTH domain containing 2 (*ythdc23*). The enhancer-like region is closer to the *myoD* promoter than the *ythdc23* promoter and *ythdc23* is not expressed almost at all in neoblasts, and hence the enhancer-like region is likely linked to *myoD* rather than *ythdc2*. (**C**) The neural FSTF *nkx6-1* has a distal enhancer-like region marked in red (width 1000 bp) downstream of the gene body. (**D**) The neural FSTF *soxB1-1* has a distal enhancer-like region marked in red (width 1000 bp) upstream of the gene body.

linked to *tbx* genes (*tbx-20* 142 motifs, *tbx-2/3c* 34 motifs, *tbx-2/3a* 11 motifs) (***Figure 6A, B and D***, ***Supplementary file 2*** and). *tbx-20* had a clear intronic enhancer-like region and a distal enhancer-like region, both containing motifs for TFs implicated as positional control genes (PCGs) (hox1, hox4a, prep, smad1, smad9, zic-1, sp5) (***Molina et al., 2007***, ***Reddien et al., 2007***; ***Felix and Aboobaker, 2010***; ***Vásquez-Doorman and Petersen, 2014***; ***Tewari et al., 2019***), TFs involved in neoblast migration (snail-1 and snail-2) (***Abnave et al., 2017***), and various other TFs (***Figure 6D***).

## Elements of the planarian positional gene regulatory network are active in neoblasts

Next, we studied enhancer-like regions and ppotential regulatory links associated with the well-known planarian PCGs. Planarians are a primary model system to understand how positional information guides and directs stem cell function during regeneration (***Reddien, 2018***), and therefore some functional genomics data exists with regard to PCGs, enabling the comparison with our findings (***Tewari et al., 2019***). A constitutive positional information system is established by the regional expression pattern of PCGs that pattern the anterior-posterior (AP), dorsal-ventral (DV), and medial-lateral (ML) axes (***Reddien, 2018***).

The AP axis is patterned by the Wnt signaling pathway: high Wnt activity specifies posterior identity, while low activity specifies anterior identity (***Reddien, 2018***). Upon β-catenin knockdown, posteriorly expressed *sp5* and the Hox genes *post-2c*, *post-2d*, *lox5a*, and *hox4b* are rapidly downregulated (***Tewari et al., 2019***). In multiple vertebrate species, *sp5* is known to be a direct target of Wnt signaling (***Weidinger et al., 2005***; ***Fujimura et al., 2007***), and this regulatory link seems to be conserved in planarians (***Tewari et al., 2019***). Here, we found a putative bound *sp5* footprint in the vicinity of the *post-2c* and *lox5a* promoters, suggesting and further supporting that *sp5* regulates the expression of these posterior PCGs (***Figure 7A***). *sp5* had 431 putative bound motifs and was the fourth most bound motif in X1 cells overall, further suggesting that *sp5* mediates the broad positional information provided by Wnt signaling in planarians (***Supplementary file 2***).

Through ATAC-seq and ChIPmentation techniques, ***Pascual-Carreras et al., 2020*** screened for *cis*-regulatory elements in planarian tissues in *notum* and *wnt1* (RNAi) animals. Upon *wnt1* knockdown, posterior Hox genes *hox4b*, *post-2c*, *post-2b*, *lox5a*, *lox5b* and *wnt11-1*, *wnt11-2*, *fzd4*, and *sp5* were downregulated, replicating the results of β-catenin knockdown (***Pascual-Carreras et al., 2020***). In addition, two *foxG* binding sites were found in the first intron of *wnt1*, and *foxG* knockdown was found to phenocopy *wnt1* knockdown, supporting the hypothesis that *foxG* is an upstream regulator of *wnt1* (***Pascual-Carreras et al., 2020***). Here, we found one enhancer-like region in the first intron of *wnt1* with a high level of H3K27ac, H3K4me1, and ATAC-seq footprinting scores for Fox family TFs (***Figure 7B***). This motif implicated by footprinting analysis is the same as one of the motifs described by previous work (***Pascual-Carreras et al., 2020***). We did not see evidence of binding at the second motif, and neither was this motif within one of our predicted enhancers (***Figure 7B***).

Although the planarian Hox genes are expressed in a regionalized manner along the AP axis, the knockdown of the genes apart from *post-2d* does not result in homeostatic or regeneration-associated phenotypic changes (***Currie et al., 2016***; ***Arnold et al., 2021***). Instead, the five Hox genes *hox1*, *hox3a*, *hox3b*, *lox5b*, and *post2b* have been shown to be involved in asexual reproduction by regulating fission at potential cryptic segment boundaries (***Arnold et al., 2021***). Here, we did not find any predicted bound motifs implicated by footprinting associated with *hox1*, *hox3a*, and *hox3b* (***Supplementary file 2***), and only a few at the promoters of *lox5b* and one at *post2b* (***Supplementary file 2***, ***Figure 7—figure supplement 1A and B***). The proportional expression values of these Hox genes in X1 cells are 4% (hox1), 3% (hox3a), 63% (hox3b), 8% (lox5b), and 31% (post2b). Altogether,

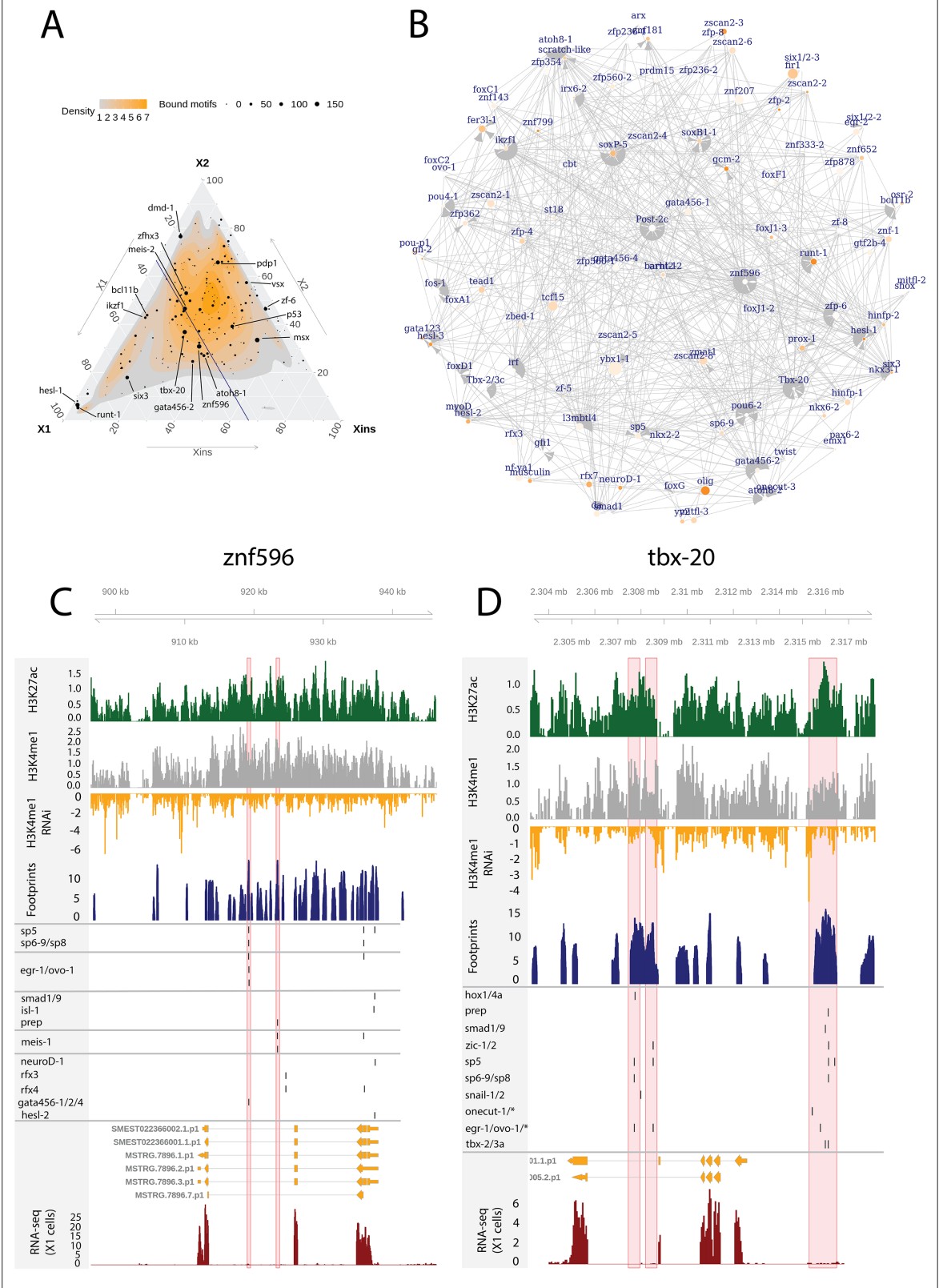

**Figure 6.** Enhancer-like regulatory regions in relation to unstudied planarian transcription factors. (**A**) Ternary plot of the proportional expression of all transcription factors (TFs) in the cell compartments (X1, X2, and Xins), and the number of bound motifs linked to the TFs. The size of the dot represents the number of bound motifs. The TFs with most bound motifs in enhancer-like regions are named and marked. (**B**) The putative gene regulatory network (GRN) of all TFs in the X1 compartment. The GRN includes all TFs with an X1 proportional expression higher than 1/3. The size of the nodes reflects the

*Figure 6 continued on next page*

*Figure 6 continued*

absolute expression in X1 cells (transcripts per million reads [TPM]), and the color of the node reflects the proportional expression in X1 cells (the more orange, the higher proportional expression). The genomic tracks of *znf596* (**C**) and *tbx-20* (**D**). The first track is the H3K27ac signal (log2 scale compared to the input sample), the second track is H3K4me1 signal (log2 scale compared to the input sample), the third track is the log2 fold-change upon *lpt(RNAi)*, the fourth track is ATAC-seq footprint score (TOBIAS footprint score), the fifth track represents bound motifs in an enhancer-like region (as predicted by TOBIAS), the sixth track represents the transcripts of the gene in the new annotation, and the seventh track represents the gene expression level of the gene in X1 cells (RNA-seq fragments per kilobase of transcript per million [ FPKM] mapped reads). (**C**) Genomic track of *znf596*. Two intronic enhancer-like regions are marked in red (width 500 bp). (**D**) Genomic track of *tbx-20*. Two intronic (width 500 bp) and one distal enhancer-like region (width 1200 bp) are marked in red.

this suggests that these Hox genes are not driving fission through their regulatory activity in adult stem cells.

*Prep*, *zic-1*, *isl-1,* and *foxD1* are PCGs that are expressed in the anterior pole of both intact and regenerating planarians (***Felix and Aboobaker, 2010***; ***Vásquez-Doorman and Petersen, 2014***; ***Vogg et al., 2014***). Interestingly, we found that these TFs are bound to several motifs in enhancer-like regions of X1 cells (***Supplementary file 1*** enhancers, *prep* 132 motifs, *zic-1/2* 330 motifs, *isl-1* 42 motifs, *foxD1/2/3* 42 motifs) and a few motifs were found to bind to the enhancer-like regions linked to *prep*, *zic-1*, *isl-1,* and *foxD1* (***Figure 7C***, ***Supplementary file 2***, ***Supplementary file 5***, and ***Supplementary file 6***). *Prep* has enhancer-like regions that have bound motifs of smad factors (Bmp signaling components) and *nkx-2-2/3/4* (***Figure 7C***). Knockdown of *nkx2-2* (also known as *DTH-1* or *nkx2.2*) causes blastemal defects both at the anterior and posterior ends, while knockdown of *prep* leads to defects in the anterior, suggesting that *nkx2-2* might work upstream of *prep* at the anterior end (***Felix and Aboobaker, 2010***; ***Forsthoefel et al., 2020***).

Lastly, we studied the Hedgehog signaling receptor *patched-1* (*ptch-1*) and found an intronic enhancer-like region containing multiple bound motifs, including the Hedgehog pathway TFs *gli-1/2* as would be expected, for the Wnt effector *sp5*, the Hox genes *lox5a/b*, BMP signaling components (*smad4-1*, *smad1/9*), the anterior pole TFs *prep* and *zic-1*, the cell migration factors *snail-1/2*, and the muscle TFs *myoD* and *twist* (***Figure 7D***). Hedgehog signaling in planarians is known to have pleiotropic functions, including interaction with the Wnt signaling pathway and glial cell function (***Rink et al., 2009***, ***Yazawa et al., 2009***, ***Wang et al., 2016***). Taken together, we found that elements of positional control gene regulatory network are active in planarian S/G2/M neoblasts and an integral part of the overall GRN determining neoblast behavior (***Figure 7E***).

## Available transcriptomic data supports putative gene regulatory networks

As an independent test of the regulatory links between TFs predicted by our GRNs, we used transcriptomic data from published RNAi knockdown experiments for TFs that had predicted footprints. We analyzed transcriptomic data for *coe(RNAi)* (***Cowles et al., 2014***), *foxD1(RNAi)* (***Vogg et al., 2014***), *pax2/5/8-1* (***Stower's Institute, 2016***), myoD(RNAi) (***Scimone et al., 2017***), nkx1-1 (***Scimone et al., 2017***), *hox1*, hox3a, and lox5b (***Arnold et al., 2021***). These available datasets had been generated from experiments using whole worms or blastemal tissues rather than the stem cell population. Nonetheless, in all cases both stem cells (X1 and X2), recent stem cell progeny (X2), and differentiated tissues (Xins) will have been present in these samples. We performed differential gene expression (DGE) analysis (***Supplementary file 7***) and obtained differentially expressed transcription factors (DETFs) as an independently derived set of functionally validated potential target TFs (***Supplementary file 8***).

To investigate whether the predicted regulatory links of our putative GRNs were supported by this collection of RNAi knockdown transcriptomic data, we compared the frequency of observed DETFs with predicted regulation by the RNAi-targeted TF in our GRNs to the expected frequency with random assignment (***Figure 8A–C***). The null hypothesis assumes that differential expression and enhancers are randomly assigned to genes and that footprints are randomly assigned to the enhancers (for details, see 'Materials and methods'). On average across these eight TFs with RNAi transcriptome data, we discovered 10.38 times more regulatory links between TFs than expected supported by transcriptome data with unbound footprints in a predicted enhancer. For TFs with predicted bound footprints, we found on average 21.09 times more regulatory links than expected (unbound p-value<0.0001,

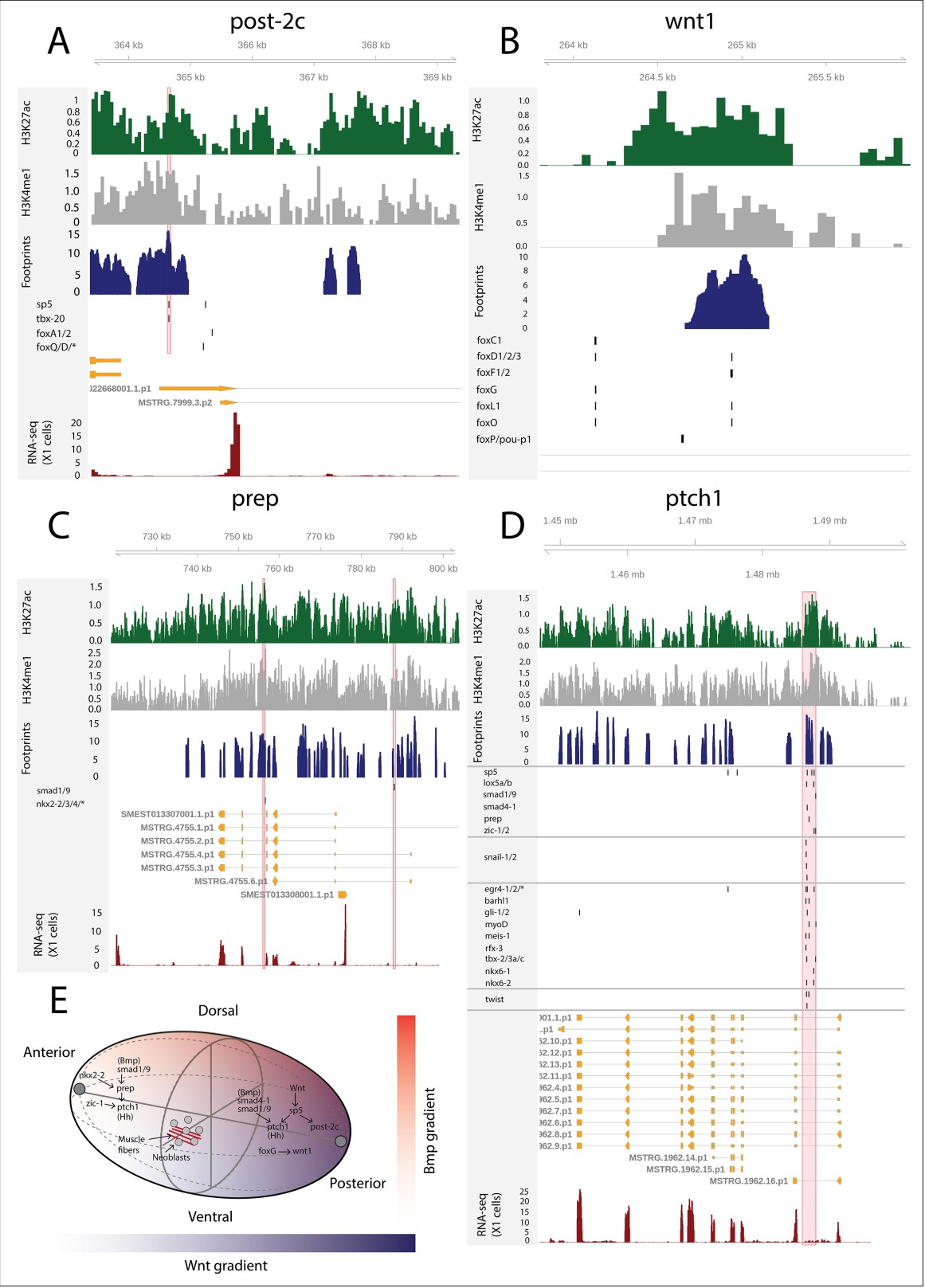

**Figure 7.** The genomic tracks of *post-2c* (**A**), *wnt1* (**B**), *prep* (**C**), and *ptch1* (**D**). The first track is the H3K27ac signal (log2 scale compared to the input sample), the second track is H3K4me1 signal (log2 scale compared to the input sample), the third track is ATAC-seq footprint score (TOBIAS footprint score), the fourth track represents bound motifs in an enhancer-like region (as predicted by TOBIAS), the fifth track represents the transcripts of the gene in the new annotation, and the sixth track represents the gene expression level of the gene in X1 cells (RNA-seq fragments per kilobase of transcript per

*Figure 7 continued on next page*

*Figure 7 continued*

million [FPKM] mapped reads). (**A**) Genomic track of *post-2c*. One upstream *cis*-regulatory regions is marked in red (width 50 bp). The foxQ/D* footprint represents motif MA0851.1 corresponding to TFs foxQ/D, foxJ1-1, foxJ1-2, foxJ1-3, foxJ1-4, foxJ1-5, foxN2/3-1, foxN2/3-2, and foxN2/3-3. (**B**) Genomic track of *wnt1*. One intronic enhancer-like region with one potential Fox family TF binding motif with evidence of binding from footprinting analysis, other Fix TF motifs lie outside the enhancer predicted by ChIP-seq data (*Pascual-Carreras et al., 2020*). (**C**) Genomic track of *prep*. Two intronic enhancer-like regions are marked in red (width 500 bp). (**D**) Genomic track of *ptch1*. One intronic enhancer-like region (width 2000 bp) is marked in red. (**E**) Schematic of position control gene (PCG) regulatory links discussed.

The online version of this article includes the following figure supplement(s) for figure 7:

**Figure supplement 1.** Regulatory regions in relation to Hox genes.

bound p-value<0.0001; *Figure 8A–C*). Results varied between knockdown experiments, indicating that the whole worm transcriptomic data reflects stem cell transcriptomics better for some TFs than others (*Figure 8A–D*). The predictive power of the GRNs developed in our study correlates with the proportional expression of the knockdown TF in the X1 compartment, which only contains stem cells, and inversely correlates with the level of expression in the Xins compartment, which only contains differentiated cells (*Figure 8D*, *Figure 8—figure supplement 1A and B*), suggesting that our GRNs are indeed reflective of activity in stem cells but not necessarily in postmitotic differentiating or differentiated cells.

For instance, we examined the potential regulatory links of coe in our GRN model and found eight putative target TFs associated with an enhancer containing a potentially bound footprint for coe. Upon coe(RNAi) of whole worms, we observed eight DETFs, and by collating these two datasets, we obtained pou4-1 as the only DETF containing a putative bound coe footprint (*Figure 8D*). This putative footprint is located within an enhancer downstream of the gene body (*Figure 8D*). Indeed, *Cowles et al., 2014* tested DETFs in their transcriptomic analysis and observed that both coe(RNAi) and pou4-1(RNAi) lead to neural defects, demonstrating that the regulatory link corroborated by our GRN could be functionally validated within the limits of the experimental toolkit of planarians. As for unstudied zinc fingers, we found that zfhx3 is differentially expressed in lox5b(RNAi) whole worms and has a putative bound footprint of lox5b within an enhancer upstream of the gene body (*Figure 8E*). These analyses overall independently validate some of our predictions and provides a set of high-confidence predictions for further studies of TFs that were found to be differentially expressed after RNAi of TFs predicted to bind a nearby enhancers.

## Discussion

In this study, we improved the current genome annotation of *S. mediterranea* by integrating available RNA-seq data and identified new coding and non-coding transcripts. We reviewed the literature on both computationally and experimentally derived planarian TFs and defined a high-confidence set of TFs. If possible, we also predicted binding motifs for the planarian TFs. We developed and performed ATAC-seq on the proliferating stem cell compartment and determined genomic regions of open chromatin. We analyzed genome-wide profiles of the histone modifications H3K27ac and H3K4me1, along with chromatin accessibility data, and used these epigenetic datasets to delineate putative active enhancers in planarian stem cells. Lastly, we identified TFs binding to the enhancers of potential target genes and constructed hypothetical GRNs active in the stem cells.

We found more than 50,000 new isoforms to known transcripts in our annotation. This information has already helped to improve the clustering of single-cell RNA-seq data (*García-Castro et al., 2021*). Our approach demonstrated that the principle of integrating available RNA-seq data into a comprehensive expression-driven annotation can significantly improve genome annotation. We were also able to annotate more than 7000 new loci, with over 3000 predicted to be coding, increasing the number of protein coding genes by more than 10%. These new proteins tend to be shorter, are more likely to be proteins specific to planarians, but are expressed at similar levels to the rest of the coding transcriptome. This, together with the extensive annotation of alternate splice forms, gives a more complete picture of the genome of the model planarian.

Altogether, we annotated 551 TFs that were distributed evenly with respect to enrichment across all three FACS cell compartments in planarians (X1, X2, and Xins). In planarians, most studied functionally have been involved in regulating differentiation. Here, we define many neoblast-enriched

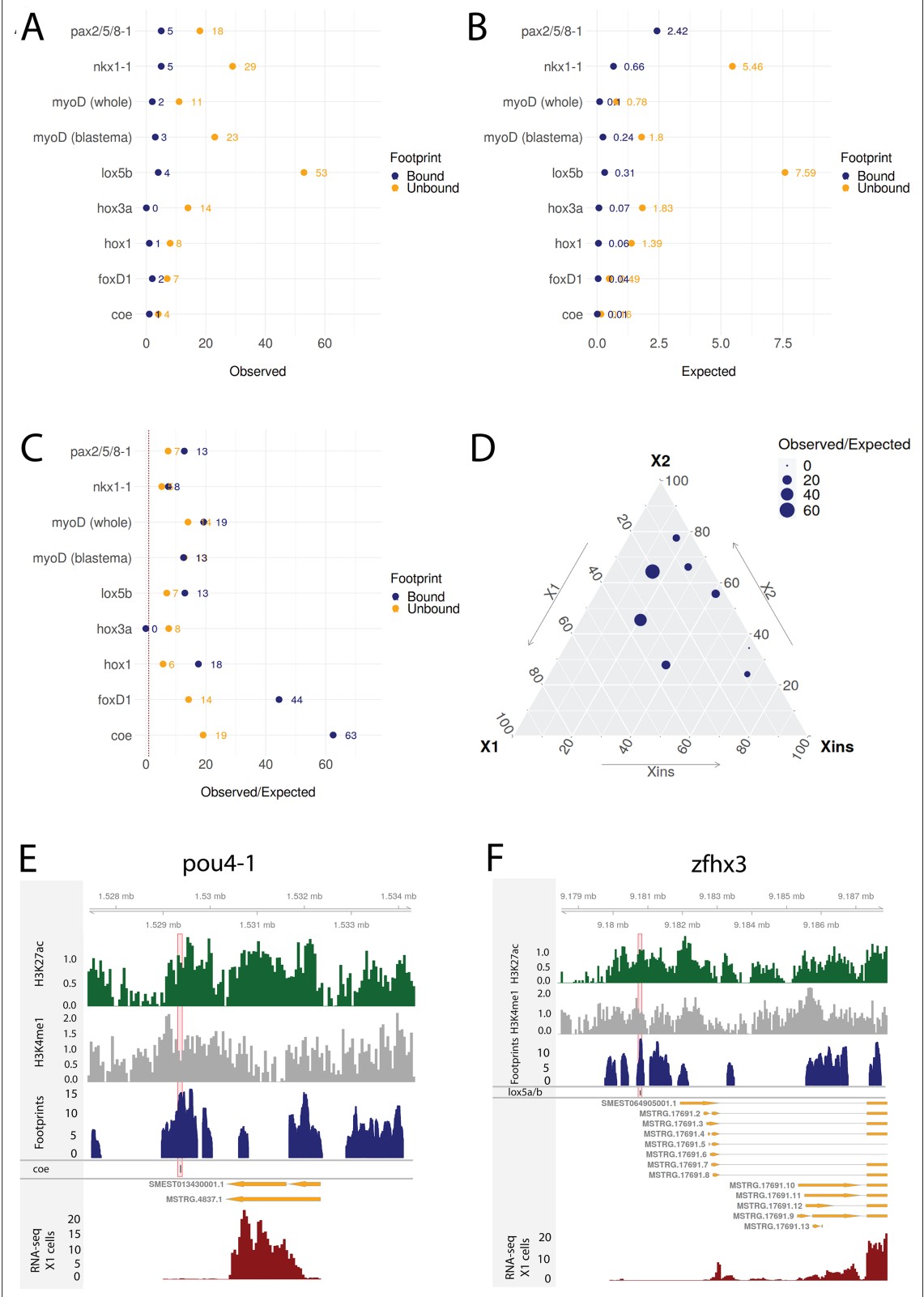

**Figure 8.** Transcriptomic data from knockdown experiments supports predicted regulatory links. (**A**) The number of observed regulatory links between the knocked down transcription factor (TF) (y-axis) and differentially expressed TFs in our gene regulatory network (GRN) model. The regulatory links with unbound and bound footprints are displayed separately. (**B**) The number of expected regulatory links between the knocked down TF (y-axis) and differentially expressed TFs. (**C**) The ratio of observed and expected regulatory links as obtained from transcriptomic data for knocked down TFs. The

*Figure 8 continued on next page*

*Figure 8 continued*

regulatory links with unbound and bound footprints are displayed separately. (**D**) A ternary plot displaying the observed/expected ratio alongside the proportional expression values of the X1, X2, and Xins compartments. The size of the dot corresponds to the observed/expected ratio. The ratio is higher closer to the X1 and X2 compartment maxima, indicating that the GRN model predicts links better for TFs with higher expression in stem cells. (**E**) Genomic track of pou4-1. One coe bound footprint within a downstream enhancer-like region (width 100 bp) is marked in red. (**F**) Genomic track of zfhx3. One lox5a/b bound footprint within an upstream enhancer-like region (width 100 bp) is marked in red.

The online version of this article includes the following figure supplement(s) for figure 8:

**Figure supplement 1.** Correlation of compartment-specific proportional expression and the enrichment of observed regulatory links.

TFs that have not been formally studied, with potential roles in maintaining neoblast pluripotency and potentially early lineage commitment. A future systematic functional screen of these may help to uncover the GRN network that maintains pluripotency analogous to that in mammals and vertebrates (*Takahashi and Yamanaka, 2006*; *Takahashi et al., 2007*). In particular, we found a large number of uncharacterized planarian zinc finger TFs, and this large and diverse family of TFs is still relatively poorly studied, perhaps because they evolve relatively rapidly in metazoans in general and often homology cannot be confidently assigned across phyla (*Albà, 2017*; *Cassandri et al., 2017*; *Najafabadi et al., 2017*). Currently little is known about transcriptional control of pluripotency in adult stem cells across animals, outside of vertebrates. In planarians, both the NuRD (*Jaber-Hijazi et al., 2013*) and BRG/Brahma complexes *Onal et al., 2012* have been implicated in regulating pluripotency through their RNAi phenotypes that block differentiation without affecting stem cell maintenance/self-renewal. By analogy with vertebrates, these chromatin remodeling complexes may directly regulate the activity of pluripotency TFs, which may include some unstudied TFs in our predicted GRNs.

Overall, for the main goals of this study, our analysis and identification of TFs allowed us to confidently assign likely binding motifs to just under half of the annotated planarian TFs (*Figure 2E*, *Supplementary file 1*). In the future as the number of planarians studies using ATAC data increases, and by looking at the actual sequence motifs implicated by footprinting analyses, it should be possible to refine motifs for planarian TFs and even define motifs for some TFs to which motifs could not be assigned, for example, in the diverse zinc finger TF group.

Based on combining epigenomic experiments and data types, we could identify putative intergenic and intronic enhancers in the planarian the genome of proliferating stem cells. The combined use of ChIP-seq data, RNAi of a histone methyltransferase combined with ChIP-seq, ATAC-seq data, and footprinting analyses together provided strong evidence for the identification of bona fide planarians stem cell enhancers. In the future, genome-wide ChIP-seq against the transcriptional cofactor p300 could serve as a complementary high-throughput approach to further substantiate these enhancers (*Visel et al., 2009*; *Schwaiger et al., 2014*), if available antibodies recognize the planarian ortholog of this protein (*Fraguas et al., 2021*; *Stelman et al., 2021*).

Verification of the function and targets of planarian enhancers is currently not possible using traditional approaches as no transgenic reporter technologies enabling enhancer-reporter constructs to be assayed are available. Therefore, we are bound to rely on less direct evidence from genome-wide sequencing technologies. Here, we assigned enhancers to target genes based on distance and the expression of the target gene, similar to established protocols used by others (*Duren et al., 2017*). We find cases where regulatory interactions suggested by previous expression-based studies of neoblast fate control and RNAi-based studies of gene function are supported by the GRNs interactions uncovered by our analyses. We find evidence that regulators of one differentiation lineage bind the enhancers of genes that regulate another, presumably acting as repressors. Our predicted GRNs were statistically supported by differential expression analyses of RNA-seq sets collected after RNAi of specific TFs (*Figure 8*), suggesting that many of the predicted regulatory interactions may be real. Experiments based on the predictions from our data combining RNAi against TFs with RNA-seq and ATAC-seq approaches will allow these GRNS to be studied further to help us understand how stem cells drive regeneration and homeostasis in planarians. In the future, both promoter-capture HiC and the co-accessibility of putative enhancers and promoters in scATAC-seq data would offer further computational possibilities to study these promoter–enhancer interactions (*Schoenfelder et al., 2015*; *Pliner et al., 2018*).

Taken together, our definition of enhancers in stem cells genome wide creates a foundation for constructing detailed GRNs to help understand regenerative mechanisms driven by stem cells in planarians.

# Materials and methods
## Reference assembly and annotations

The sexual genome SMESG.1 genome, the SMESG high-confidence annotation (SMESG-high release 1), and the SMESG annotation filtered for repeats (SMESG-repeat release 2) were downloaded from PlanMine (*Brandl et al., 2016*; *Grohme et al., 2018*). Available RNA-seq datasets were aligned to the genome with HISAT2 (version 2.1.0) using default parameter settings and providing known splice sites from the SMESG-high annotation (*Kim et al., 2015*). Transcripts were assembled and merged from the alignments with StringTie using the SMESG-high annotation as a reference (*Pertea et al., 2015*; *Pertea et al., 2016*).

The new expression-driven annotation (*García-Castro et al., 2021*, https://github.com/jakke-neiro/Oxplatys/raw/gh-pages/Schmidtea_mediterranea_Oxford_v1.gtf.zip) was compared to the SMESG-high annotation with gffcompare (*Pertea et al., 2015*; *Pertea et al., 2016*). Transcripts labeled with the class code "=" were classified as full matches corresponding to the transcripts in the SMESG.1 genome, while transcripts labeled with the class codes "c", "k", "m", "n", "j", "e", and "o" were classified as isoforms. Lastly, transcripts labeled with class codes "u" (unknown or intergenic), "i" (fully contained within a reference intron), and "x" (exonic overlap on the opposite strand) were selected as potential candidates for new coding and non-coding transcripts (*Pertea et al., 2016*; *Wu et al., 2016*, *Azlan et al., 2019*). Transdecoder was used to identify ORFs in these new transcripts, and transcripts longer than 100 amino acids were classified as putative coding transcripts (*Haas et al., 2013*; *Wu et al., 2016*, *Azlan et al., 2019*). Subsequently, InteProScan and BlastX against UniProt were used to look for protein-like structures in the translated ORFs of the remaining transcripts, and transcripts without any hits were retained as putative non-coding transcripts *Jones et al., 2014*, *Wu et al., 2016*.

## Expression values, homology, and GO

To obtain FACS-specific expression values, the same FACS RNA-seq datasets used by *Dattani et al., 2018* were used. The RNA-seq datasets were pseudo-aligned with Salmon using selective alignment, k-mer size 31 and 50 bootstrap iterations (*Patro et al., 2017*). The abundance estimates were converted to the Kallisto compatible format with wasabi, and the transcript counts were normalized with Sleuth (*Pimentel et al., 2017*). The TPM values of individual transcripts were summed to calculate TPM values for each gene. The mean TPM value across all samples in a FACS cell compartment was calculated for each gene to get the final absolute TPM value. The proportional expression values ($Xk\%$) were calculated by dividing the compartment-specific TPM value by the sum of TPM values of all compartments:

$$Xk\% = \frac{Xk(TPM)}{X1(TPM)+X2(TPM)+Xins(TPM)}, \ Xk \in \{X1, \ X2, \ Xins\}$$

Transcripts were categorized into FACS enrichment categories as follows: X1 when X1% > 0.5, X2 when X2% > 0.5, Xins when Xins% > 0.5, X1 and X2 when X1% + X2% > 0.75 and neither enriched in X1 nor X2, X1, and Xins when X1% + Xins% > 0.75 and neither enriched in X1 nor Xins, X2, and Xins when X2% + Xins% > 0.75 and neither enriched in X2 nor Xins, and ubiquitous when not enriched in any of the above categories (*Dattani et al., 2018*). Transcript lengths and expression values were compared with Kruskal–Wallis test. Ternary plots and kernel density estimation of the proportional expression values were generated with the package ggtern (*Hamilton and ggtern, 2018*). Information content (IC) was defined to represent as a singular scalar the divergence of a combination of proportional expression values from the even distribution point (X1% = X2% = Xins%):

$$IC = \ 0.01 * p_1 * \log(p_1) + 0.01 * p_2 * \log(p_2) + 0.01 * p_3 * \log(p_3)$$

where $p_1 = X1\%$, $p_2 = X2\%$, $p_3 = Xins\%$, and $\min(\{p_1, \ p_2, p_3\}) = 0.01$, meaning that proportional expression values below 0.01 were assigned the value 0.01.

The homology of the transcripts was investigated by using Blastn (*Altschul et al., 1990*). The non-coding and coding transcripts were aligned to the transcriptomes of humans (GRCh38.p13) and *D. japonica* (assembled in *García-Castro et al., 2021*). The threshold for non-coding transcripts was e-value = $10^{-5}$ and for coding transcripts e-value = $10^{-10}$. GO enrichment analysis of new coding transcripts with respect to known coding transcripts was performed using topGO with Fisher's exact test (*Alexa and Rahnenfuhrer, 2020*). The data for proportional expression and homology assigned by BLAST of all transcripts is available at https://jakke-neiro.github.io/Oxplatys/; *Neiro, 2020*.

## Identification and characterization of transcription factors

We conducted a systematic domain annotation of all transcripts by using the InterProScan resource (*Jones et al., 2014*). Transcripts with an InterProScan description 'transcription factor', a Pfam hit to the Pfam families listed in the TF database DNA-binding domain (DBD) v2.0, or a SUPERFAMILY hit listed in the SUPERFAMILY families in DBD were classified as TFs (*Wilson et al., 2008*). We used Blastn to align the potential planarian TFs to TFs in humans and fruit flies, and potential TFs without hits were filtered out. The planarian literature was systematically reviewed (*Supplementary file 1*), and if a TF was mentioned, the GenBank accession number or primer information was retrieved to establish the exact sequence in the literature. This sequence from the literature was aligned to the potential TFs (default parameters for GenBank accession entries and word size 10 for primers) to correctly assign the TFs in the literature to our transcripts (*Supplementary file 1*). If a planarian TF had previously been described in the literature under a certain name, one name was chosen as the primary name, while all other alternative synonyms used for the same TF were listed as secondary names (*Supplementary file 1*). If a potential TF had not been described in the planarian literature, the name of the best human or fruit fly Blast hit was used (*Supplementary file 1*). The TFs were categorized into four main groups (Basic domain, Zinc fingers, Helix-turn-helix, Alpha-helical, Immunoglobulin-fold, Beta-hairpin, Beta-sheet, Beta-barrel, Others) according to the TRANSFAC database (*Stegmaier et al., 2004*; *Supplementary file 1*).

Ternary plots, kernel density estimation, and the information content of the proportional expression values of TFs were calculated as for other transcripts (see 'Expression values, homology, and GO'). Hierarchical clustering of FACS proportions was performed using Euclidean distance and Ward's method with the package hclust (*van den Boogaart and Tolosana-Delgado, 2008*). The frequencies of different TF domains among the cellular enrichment classes X1, X2, and Xins (for which FACS proportional expression is higher or equal to 50%) were compared with the chi-square test using Yates correction.

## Motifs of transcription factors

The motifs of the TFs (*Supplementary file 1*) were predicted by using the JASPAR profile inference tool (*Fornes et al., 2020*). The protein sequences were retrieved with Transdecoder (see 'Reference assembly and annotations'). The information content for each motif was determined by calculating the mean of Shannon's entropy at each position in the position weight matrix (PWM). The motifs were visualized with seqLogo (*Bembom, 2019*).

## ATAC-seq library preparation

A standard protocol was used for preparing the ATAC-seq library (*Buenrostro et al., 2013*). The X1 cell compartments were isolated by FACS (*Romero et al., 2012*), and two replicates with 120,000–250,000 cells were collected from each compartment. The cells were washed and centrifuged (1 XPBS, 1200 RPM). Lysis buffer (10 mM Tri–Cl [pH 7.5], 10 mM NaCl, 3 mM $MgCl_2$, 0.1% NP-40) was added, the cells were centrifuged (500 RPM, 10 min, 4°C), and the nuclei pellet was collected. Then, the transposase mix (25 μl 2X TD Buffer, 2.5 μl Tn5 Transposase, 22.5 μl of nuclease-free water) was added, and the cells were resuspended and incubated (37°C, 60 min). Finally, DNA was isolated using the Zymogen Clean & Concentrator Kit, and eluted in EB buffer.

Subsequently, the eluted DNA was used for PCR amplification and library purification. DNA was amplified using a standard reaction (10 μl of purified transposed DNA, 10 μl of nuclease-free water, 15 μl Nextera PCR Master Mix, 5 μl of PCR primer cocktail, 5 μl Index Primer 1, 5 μl Index Primer 2, 72°C, 3 min, 98°C, 30 s, 14 cycles of 98°C, 10 s, 63°C, 30 s, 72°C for 1 min). Finally, the libraries were cleaned with the AMPure bead purification kit. Samples were paired-end sequenced on the

Illumina NextSeq. The sample replicates can be found on Sequence Read Archive (SRR18923214 and SRR18923213).

## ChIP-seq library preparation and sequencing

The H3K27ac ChiP-seq library was prepared and sequenced using established protocols (*Mihaylova et al., 2018*; *Dattani et al., 2018*). A total of 600,000–700,000 planarian X1 cells were isolated for each experimental replicate (actual sample and input control). Two sample replicates (Sequence Read Archive SRR18925505 and SRR18925504) and two input replicates (Sequence Read Archive SRR18925503 and SRR18925502) were prepared. The H3K27ac Abcam ab4729 antibody was used for immunoprecipitation.

## ChIP-seq data analysis

The ChIP-seq data with respect to H3K4me1 and H3K4me1 after *lpt(RNAi)* was reanalyzed (*Mihaylova et al., 2018*, PRJNA338116) alongside the H3K27ac data prepared here. The reads were quality-checked with FastQC and trimmed with Trimmomatic (*Andrews et al., 2010*; *Bolger et al., 2014*). The reads were aligned to the SMESG.1 genome with Bowtie2 (*Langmead et al., 2009*). Only uniquely mapped reads and reads with a quality score greater than 10 were retained with Samtools (*Li et al., 2009*). Peaks were called with MACS2 with default parameters for H3K4me1 and the broad peak parameter for H3K27ac (*Zhang et al., 2008*). Coverage tracks were generated with deepTools bamCoverage, and the mean coverage track was determined with wiggletools (*Zerbino et al., 2014*; *Ramírez et al., 2016*). Heatmaps of ChIP-seq profiles were generated with deepTools computeMatrix and plotHeatmap 1000 bp upstream and downstream of the peak center (*Ramírez et al., 2016*). The mean peak value was calculated as the mean of the ChIP-seq signal 1000 bp upstream and downstream of the peak center. The H3K27ac peaks that were at most 500 bp from the nearest H3K4me1 peak were selected as putative enhancers. Random genomic regions were generated with bedtools random (*Quinlan, 2014*). The difference of mean peak value at enhancers and random regions was tested with Wilcoxon's test. Detailed analysis and code are available at https://jakke-neiro.github.io/Oxplatys/; *Neiro, 2020*.

## ATAC-seq data analysis and motif footprinting

The quality of paired-end reads was assessed with FastQC and adapter sequences were removed with Trimmomatic (*Andrews et al., 2010*; *Bolger et al., 2014*). The reads were aligned to the SMESG.1 genome with Bowtie2 (*Langmead et al., 2009*). Only uniquely mapped reads and reads with a quality score greater than 10 were retained with Samtools (*Li et al., 2009*). Duplicates were removed with Samtools (*Li et al., 2009*). Insert sizes were calculated with Picard CollectInsertSizeMetrics (http://broadinstitute.github.io/picard/; *Magner, 2022*).

For coverage track generation and peak calling, samples were downsampled to a standard of ~20M reads using Picard DownsampleSam (http://broadinstitute.github.io/picard/; *Magner, 2022*). Deeptools2 bamCoverage was used to generate coverage tracks for each sample in bigwig and bedgraph formats by normalizing with respect to reads million mapped reads (RPKM) (*Ramírez et al., 2016*). TOBIAS was used to look for footprints in the ATAC-seq by using the motifs predicted by JASPAR (see 'Motifs of transcription factors'; *Bentsen et al., 2020*). BAM files of mapped reads are available at https://jakke-neiro.github.io/Oxplatys/; *Neiro, 2020*.

## Enhancer targets and gene regulatory networks

The putative enhancers were assigned to the nearest TSS with ChIPSeeker (*Yu et al., 2015*). The footprints in enhancers were used to create links between TFs and target genes for network construction using a Python script (available at https://jakke-neiro.github.io/Oxplatys/; *Neiro, 2020*). Networks were visualized with the R package Igraph (https://github.com/igraph/rigraph; *Nepusz, 2022*) while the genomic regions were visualized with Gviz (*Hahne and Ivanek, 2016*).

## Functional transcriptomics and validation of gene regulatory network predictions

To calculate the odds ratio of our approach, we defined the expected value of the number regulatory links for a given TF. A regulatory link for a given TF assumes that the target gene is linked to an

enhancer and that the enhancer contains a footprint for the given TF. In addition, if we assume that all differentially expressed genes are real direct target genes for the given TF, then the target gene of the regulatory link is also differentially expressed upon perturbation of TF activity. First, we define $A$ as the event that a TF is differentially expressed. The probability of this event is

$$P\left(A\right) = \frac{n_{dge}}{n_{ge}}$$

where $n_{ge}$ is the total number of TFs in the genome, and $n_{dge}$ is the number of differentially expressed TFs. Second, we define $B$ as the event that a given TF is linked to an enhancer. The probability of this event is

$$P\left(B\right) = \frac{n_{cis}}{n_{ge}}$$

where $n_{cis}$ is the total number of TFs linked to an enhancer. Third, we define $C$ as the event that an enhancer contains a footprint (bound motif) of the given TF. The probability of this event is

$$P\left(C\right) = \frac{n_{cisTF}}{n_{cis}}$$

where $n_{cisTF}$ is the number of enhancer-linked genes containing the footprint of the given TF in their enhancers.

We assume that the events $A$, $B$, and $C$ are independent. The expected number of differentially expressed genes linked to enhancers with the footprint of a given TF is equal to

$$P\left(A \cap B \cap C\right) n_{ge} = P\left(ABC\right) n_{ge} = P\left(A\right) P\left(B\right) P\left(C\right) n_{ge}$$

## Code availability

All codes are available in a notebook provided as a zip file (*Source code 1*).

## Additional information

### Funding

| Funder | Grant reference number | Author |
|---|---|---|
| Medical Research Council | MR/T028165/1 | Aziz Aboobaker |
| Biotechnology and Biological Sciences Research Council | BB/J014427/1 | Anish Dattani |
| Biotechnology and Biological Sciences Research Council | BB/M011224/1 | Jakke Neiro |

The funders had no role in study design, data collection and interpretation, or the decision to submit the work for publication.

### Author contributions

Jakke Neiro, Formal analysis, Validation, Investigation, Visualization, Methodology, Writing - original draft, Writing - review and editing; Divya Sridhar, Investigation, Methodology, Writing - review and editing; Anish Dattani, Formal analysis, Investigation, Visualization, Methodology, Writing - review and editing; Aziz Aboobaker, Conceptualization, Supervision, Funding acquisition, Investigation, Methodology, Writing - original draft, Project administration, Writing - review and editing

### Author ORCIDs

Jakke Neiro (iD) http://orcid.org/0000-0002-5077-4958
Anish Dattani (iD) http://orcid.org/0000-0003-2211-1521
Aziz Aboobaker (iD) http://orcid.org/0000-0003-4902-5797

### Decision letter and Author response

Decision letter https://doi.org/10.7554/eLife.79675.sa1
Author response https://doi.org/10.7554/eLife.79675.sa2

## Additional files

### Supplementary files

• Supplementary file 1. Full list of transcription factors (TFs) annotated in the *S. meditteranea* genome. TFs are named according to existing names in the literature or according to the currently agreed convention. The absolute (transcripts per million) or proportional expression in the fluorescence-activated cell sorting (FACS)-associated cell compartments X1, X2, and Xins is also recorded.

• Supplementary file 2. Full list of putative enhancers in adult stem cells defined by ChIP-seq and ATAC-seq analyses. Includes the position in the genome and the identify of the predicted target gene based on proximity to the predicted transcriptional start site.

• Supplementary file 3. Full list of predicted ATAC-seq footprints in the genome, including predicted transcription factor (TF)/TF family, position of binding site in the genome, and position of the peak/ enhancer in the genome.

• Supplementary file 4. Full list of motifs with and without footprints organized by transcription factor (TF)/TF family in predicted enhancer regions.

• Supplementary file 5. Overview of number of enhancers, predicted bound motifs (footprints), and unbound motifs for fate-specific transcription factors (FSTF) expressed in adult stem cells.

• Supplementary file 6. Full list of transcription factors (TFs), involved in a putative stem cell gene regulatory network (GRN). Includes TF with expression >1/3 in the X1 compartment and includes other TFs implicated in regulating or being regulated by these TFs.

• Supplementary file 7. Differential expression analysis of transcription factor (TF) RNAi with RNA-seq datasets.

• Supplementary file 8. Differentially expressed transcription factor (TFs) after TF RNAi.

• MDAR checklist

• Source code 1. Full set of codes used for analyses in this study.

### Data availability

Data and analyses are available at https://jakke-neiro.github.io/Oxplatys, (copy archived at swh:1:rev:51a4d412daf02503691c903cd64a6f6e78dc1b25). All analysis code is provided in supplementary file New sequence data are available at the NCBI under Bioproject ID PRJNA832235, https://www.ncbi.nlm.nih.gov/bioproject/832235.

The following dataset was generated:

| Author(s) | Year | Dataset title | Dataset URL | Database and Identifier |
|---|---|---|---|---|
| Neiro J, Sridhar D, Dattani A, Aboobaker A | 2022 | Identification of enhancer-like elements defines regulatory networks active in planarian adult stem cells | https://www.ncbi.nlm.nih.gov/bioproject/832235 | NCBI BioProject, PRJNA832235 |

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
