## [Editor Report]

The authors have generated a comprehensive list of transcription factors in the planarian *Schmidtea mediterranea*, providing insight into which factors are highly expressed in the stem cell compartment. The computational identification of transcription factors and putative enhancers will be helpful to researchers studying stem cell and regenerative biology using planarians and provides a large dataset that informs upon the evolution of regulatory sequences and transcription factor function.

---

## [Decision Letter]

[Editors' note: this paper was reviewed by Review Commons.]

---

## [Author Response]

General Statements

The main goals of this study were to provide a closer to complete picture of the planarian genome/transcriptome and on this foundation to broadly identify enhancer elements in adult stem cells by applying functional genomic approaches to this cell population. Currently there is no rigorous attempt to this, with multiple lines of evidence, in an adult stem cell population outside of the vertebrate lineage. We perform a set of genome wide analyses that support the identification of regions that are most probably active enhancers in stem cells. At this stage we do not propose to extend this study to specific individual transcription factors or enhancers as we think this lies beyond the scope of this study. Publication of our study at this stage establishes this as an opportunity for all planarian research group, not just ours. The evidence we provide to identify enhancers is sufficient based on similar genome wide studies predicting enhancers in other models, and the assignment of enhancer status conservative enough to give confidence to our set of predictions. We note one key piece of ‘functional’ data worth highlighting is that RNAi of the methyltransferase responsible for H3K4me1 at enhancers leads to reduction of this mark at our predicted enhancers, but not elsewhere in the genome (Figure 3K). We certainly agree that each individual predicted enhancer and regulatory interaction of interest needs to be studied for its exact role in stem cells and/or during regeneration and will make sure the revision of the text reflects this.

We have added extensive new analyses to address two of the reviewers’ comments that further functional evidence was required through new experiments using RNAi of TFs coupled to RNA-seq, to find independent evidence of for links between TFs and predicted targets. As this research project was affected by the pandemic, we were unable to currently afford this significant extension to the work in terms of lab time or resources, rather we see this as the work of many labs in the coming years. Nonetheless, we acted on the reviewers’ suggestions by using existing TF RNAi/RNA-seq data sets for TFs that we predict to be active in the stem cell regulatory networks predicted in our initial submission (new Figure 8). Despite these experiments being performed in mixed tissues (stem cells, progeny, differentiated cells) we still observe clear evidence for differential expression of TFs predicted by our enhancer and footprint predictions to be downstream of the TF targeted by RNAi. This improves the confidence of our original findings significantly and we are grateful for the reviewers’ for suggesting this approach. Overall, we strongly agree with the reviewers’ comments that our work will be a valuable resource for those studying regeneration and animal stem cells.

Reviewer #1 (Evidence, reproducibility and clarity (Required)):In this manuscript Neiro et al. aim to expand our knowledge on the regulation of gene expression in stem cells of the planarian model organism.As a first step the authors used published available data to expand the repertoire of the planaria transcriptome. By combining 183 RNAseq datasets the authors were able to identify thousands of new coding and non-coding transcripts. They then screened for TF motifs in the new annotations, identifying 551 putative TFs, of which 248 were already described in the planarian literature.The most substantial contribution of this work to the field of stem cells and planaria biology is the characterization of new putative enhancers that were identified by performing H3K27ac ChIP-seq and ATAC-seq and combining these data with previously published H3K4me1 ChIPseq dataset.

We thank the reviewer for their careful assessment of our work, we agree that the identification of likely enhancers genome wide is a substantial contribution. Equally the improved annotation of all genes, including the collected set of transcription factors we choose to focus on here, is a substantial step forward for the planarian research community.

By overlapping H3K27ac and H3K4me1the authors find 5,529 new enhancers, for which they report a higher chromatin accessibility than random points in the genome as assessed by ATAC-seq. By using ATAC-footprints Neiro et al. refined the subset of TFs that have binding motifs in the predicted enhancer-like regions and present a list of 22,489 such factors.The manuscript is well written and organized and overall, the reported data will provide an important resource to study gene expression regulation in planaria's stem cells. However, this manuscript would greatly benefit from some functional validation to support the predicted gene regulatory networks. One option would be to use a CRISPR-dCas9-KRAB system to silence the putative enhancers identified in the manuscript and check by qPCR the expression of nearby genes.

Currently mis-expression technologies, in order too directly test enhancer elements in driving expression, are still not available in planarians. This also preempts us using the suggested silencing system used in mammals and other animals with robust mis-expression tools.

If this type of experiment is not feasible in planaria (I am not an expert in this model organism) another simple but key experiment would be to perform a knockdown of one (or more) putative enhancer-bound TFs identified in this study followed by RNA-seq. This would allow the authors to verify what are the target genes of the putative enhancer-bound TFs and if they correspond to the predicted gene networks they identified. Simultaneously, this experiment would allow the authors to verify if there are any changes in the expression of differentiation/pluripotency markers as a result of the knockdown of the putative enhancer-bound TF.

These experiments are possible, but this would be the work of many labs in the future expert in studying those TFs, the organs and tissues they are expressed in, and their roles in planarian stem cells and regeneration. However, what we have done is analyze existing RNA-seq data collected after specific TF RNAi (new Figure 8). There are several studies where TF have been studied and RNA-seq performed after RNAi. Although these studies are performed in specific experimental regenerative contexts, and not specifically in stem cells, it was nonetheless possible to look at expression changes of potentially downstream TFs with predicted enhancers for those TFs knockdown by RNAi. We analyzed these data and added it to the manuscript, rather than perform further TF RNAi experiments, as this was just about feasible within a 3-month revision time. We would add that currently there are no genes implicated in controlling pluripotency in the same way we might consider, for example, OSKM in mammals. Our identification of the TFs enriched in stem cell expression and implicated in binding predicted enhancers suggests future candidates.

Our additional analyses (Figure 8) clearly show that TF predicted to be regulated by our enhancer predictions are significantly enriched amongst differentially expressed TFs after RNAi of the predicted regulating TF. We observed differences between the levels of unbound enhancers (10x enrichment) and bound enhancers with footprint evidence (21x enrichment) across all the TF RNAi/RNA-seq studies, and a correlation between the predictive power of our GRNs and the proportion of expression of the knockdown TF in the stem cell compartment. This suggests that our predicted GRNs are reflective of activity in stem cells, but not of other cells.

Minor revision:The authors have mostly focused on the identification of enhancer-bound TFs. However, it would be interesting to look at differential enrichment of TFs in promoters versus enhancers and identify if there are specific factors that are enriched specifically at the planarian newly identified enhancer regions.

We have not looked at potential TF binding sites near promoters/transcriptional start sites. We have added an analysis to look at this (new Figure S3). This analysis does not reveal any TFs or TF families that show a pattern of having binding sites that are more proximal to promoters than others. We did not pursue this further as it was beyond the scope of our study.

All tornado plots are missing a color bar (Figure 3 and FigS2)

We have fixed this error.

There is a typo in the discussion: "the combined use of ChIP-seq data, RNAi of a histone methyltransferase combines with ChIP-seq" should be changed to "combined".

We have fixed this and other typographical errors.

Reviewer #1 (Significance (Required)):The manuscript is well written and organized and overall the reported data will provide an important resource to study gene expression regulation in planaria's stem cells.

We thank the reviewer for their appreciation of our work

Referees cross-commentingI agree with the other reviewers that additional functional data should be added to support the author's claims (such as knock down of potential TFs that are identified by computational analyses and assessing the impact on gene expression).

See response above, with regard to adding further analyses , see Figure 8 and new Results section.

In addition, as noticed by the third reviewer, all data should be made publicly available to the scientific community.

We have made all data publicly available and have submitted all relevant data to public database repositories in advance of final publication.

Reviewer #2 (Evidence, reproducibility and clarity (Required)):Summary:This manuscript aims at identifying enhancers in the planarian *Schmidtea mediterranea*. The authors start with the integration of transcriptome with genome sequencing data to more precisely annotate the genome of the planarian Schmidtea mediterranea. The second part of the manuscript actually then deals with the identification of potentially active enhancer elements in adult stem cells of this regenerating organism using genomic techniques like ATAC-seq and ChIP-seq of histone marks combined with motif searches and in silico footprint analysis. Using these data, the authors predict regulatory interactions potentially critical for pluripotency and regeneration in planarian adult stem cells.Major comments:- Are the key conclusions convincing?1) The authors claim (already in the abstract) that their study identifies enhancers regulating adult stem cells and regenerative mechanisms. This is an over-statement found throughout the manuscript, as none of these enhancers are functionally tested nor is it shown that target gene expression changes when transcription factors predicted to interact with such enhancers are knocked down.

We agree and it was not our intention to overstate our results, this is why we have tried to refer to putative enhancers, enhancer-like elements etc in manuscript from the title onwards. Only once we have demonstrated a set of elements with key conserved and widely supported characteristics do we suggest we have a set of higher confidence enhancers to study. However, we have adjusted the manuscript to reflect that our claims await direct testing as is the case for all enhancers implicated with the approaches used here.

Another example is at the end of paragraph 1 of section 2.4. Here the authors claim that identifying many fate-specific transcription factor genes in the vicinity of potential enhancers is a further proof that the identified regions represent "real enhancers". It strongly supports this hypothesis, but no evidence for real enhancer activity.

We agree the total body of evidence strongly supports that we have identified enhancer elements, but have adjusted the language to suggest further directed functional work will follow from many groups.

Thus, although the authors state that the regulatory interactions and networks they predict from their data can be studied now in future, they should be more careful with their wording and correct these over-statements. Therefore, the key conclusion is that they identified by various techniques potential enhancers, which are close to genes controlling adult stem cells and potentially controlling these genes, which has to be shown by further analyses.

We agree.

Thus, also the title needs to be changed.

We have changed ‘enhancer-like’ to “putative enhancer-like” and “defines” to “predicts” in the title as well as broadly adjusting the text to caveat that further work will clarify their functions and roles.

The authors have no proof that the networks are active in planarian adult stem cells, as they do not show that the predicted networks are active in the presented way.

We agree, see comments above. It was not our attention to claim we are showing pathways that were definitively active, but rather predicted by our experiments and analyses of the data from these experiments.

2) Similarly, the identification of TF motifs within these potential motifs strongly suggests but not shows that these factors are binding, even when these sites were found to be bound by a protein using the ATAC-seq footprinting analysis. Thus, the authors need to be careful with their wording. One example is in the second paragraph of section 2.5, where the authors write that "We found that numerous FSTFs were binding to putative intronic enhancers.… ". The motif suggests that these factors bind, however, they have no experimental confirmation that these sequences are indeed bound by the planarian TFs.

We agree and have been careful to emphasize “predicted” and “putative”

In sum, this manuscript uses existing genomic tools to define potential enhancer regions in the planarian *Schmidtea mediterranea*. The manuscript is informative yet descriptive, as tit presents no functional evidence for any of the predictions. If further toned down, the key conclusions are valid.

Future functional experiments to test the roles of all TFs and enhancers is now possible due to our work. The combination of data and analyses provides strong support of enhancer elements activity in stem cells across the genome. We have added analyses of previous RNAi work on TFs which provide strong cross validation for our predicted TF GRNs (Figure 8) and RNAi of the H3 K4 mono-methyl transferase provides genome wide functional validation of the predicted enhancer regions.

– Should the authors qualify some of their claims as preliminary or speculative, or remove them altogether? The experiments performed are well designed and in line with what is known in the field about enhancer architecture. However, as this model system is not very well characterized on that level and the authors do not provide real experimental evidence that any of the identified regions has really enhancer activity and that any of the identified motifs binds indeed the predicted TF, the authors need to be very careful with their statements. The authors should maybe emphasize even stronger that all the GRNs predicted under section 2.6 are really preliminary and need to be validated.

Yes, we are happy to have been even clearer about this as the reviewer suggests

– Would additional experiments be essential to support the claims of the paper? Request additional experiments only where necessary for the paper as it is, and do not ask authors to open new lines of experimentation.One experiment that could provide more evidence for their predicted regulatory interactions is to knock-down one of the FSTFs for which motifs have been identified in potential enhancer regions and to study expression of associated genes (to confirm that the enhancers potentilla bound by these TFs control the expression of associated genes) or by analyzing the chromatin status of selected chromatin regions (by Q-PCR). These experiments would strongly support the claims of the authors. However, it also depends strongly on the journal whether I would consider these experiments essential or "nice to have".

This suggestion of possible extra experiments is very similar to that of Reviewer 1. We are copying our earlier comment as this also addresses this point.

These experiments are possible, but this would be the work of many labs in the future expert in studying those TFs and their roles in planarian stem cells and regeneration. However, what we have done is analyze existing RNA-seq data after specific TF RNAi, where this data is available. There are several studies where TF have been studied and RNA-seq performed after RNAi. Although these studies are performed in specific experimental regenerative contexts, and not specifically in stem cells, it was nonetheless possible to look at expression changes of potentially downstream TFs with predicted enhancers for those TFs knockdown by RNAi. We analyzed these data and added it to the manuscript, rather than perform further TF RNAi experiments, as this was just about feasible within a 3-month revision time. We would add that currently there are no genes implicated in controlling pluripotency in the same way we might consider, for example, OSKM in mammals. Our identification of the TFs enriched in stem cell expression and implicated in binding predicted enhancers suggests future candidates.

Our additional analyses (Figure 8) clearly show that TF predicted to be regulated by our enhancer predictions are significantly enriched amongst differentially expressed TFs after RNAi of the predicted regulating TF. We observed differences between the levels of unbound enhancers (10x enrichment) and bound enhancers with footprint evidence (21x enrichment) across all the TF RNAi/RNA-seq studies, and a correlation between the predictive power of our GRNs and the proportion of expression of the knockdown TF in the stem cell compartment. This suggests that our predicted GRNs are reflective of activity in stem cells, but not of other cells.

– Are the suggested experiments realistic in terms of time and resources? It would help if you could add an estimated cost and time investment for substantial experiments.This reviewer is not an expert in *Schmidtea mediterranea*, thus it is hard to judge how time consuming these experiments would be. Cost-wise they should be feasible, as it would include primarily Q-PCR experiments. And some functional back-up of their claims would be very helpful.

See previous comment regarding additional analyses. We have managed these just about within the 3 month revision window.

– Are the data and the methods presented in such a way that they can be reproduced?For the parts I can judge, yes.– Are the experiments adequately replicated and statistical analysis adequate?It is not clear from the manuscript how many replicates of the ChIP-seq experiments were done.

ChIP-seq replicate data (duplicates) has been explicitly added to the methods

Minor comments:– Specific experimental issues that are easily addressable.– Are prior studies referenced appropriately?For the literature I can judge, yes.– Are the text and figures clear and accurate? The figures are clear, the text (besides over-statements) is clear. However, the writing can be improved. A few examples: section 2.2 paragraph 1: "… we found 248 to be described in the planarian literature in some way." In which way described?; same paragraph: "… but significantly we could identify new homologs of.…" what does significantly mean? Which test etc? section 2.2, last paragraph: "Most TFs assigned to the X1 and Xins compartments and the least to the Χ2 compartment", "Very few TFs had expression in X1s and Xins to the exclusion of Χ2 expression as would be expected by overall lineage relationships"; what do these sentences mean?

We thank the reviewer for paying careful attention to the language in our manuscript throughout. We will provide clearer explanation of the sentences indicated. We have better explained terms specific to the planarian model system that are obviously not intuitive and be clear in the figure legends, methods and included code notebook about statistical tests.

– Do you have suggestions that would help the authors improve the presentation of their data and conclusions?No over-statements.

See previous comments agreeing with the careful adjustment of our language to avoid this.

Reviewer #2 (Significance (Required)):– Describe the nature and significance of the advance (e.g. conceptual, technical, clinical) for the field. This manuscript identifies genome-wide potential enhancers in adult planarian stem cells, and thus represents a very valuable resource for the community to study these enhancers and the gene regulatory networks they control in the future.– Place the work in the context of the existing literature (provide references, where appropriate).As I am not a planarian scientist, it is hard to judge this part.– State what audience might be interested in and influenced by the reported findings.In my opinion, this work will be primarily interesting for people working with planarian. When functional data exist, this might be also interesting for researchers working generally on regeneration.

Given the nature of our data we also think all groups working on animal stem cells would be interested in our data and analyses.

– Define your field of expertise with a few keywords to help the authors contextualize your point of view. Indicate if there are any parts of the paper that you do not have sufficient expertise to evaluate.My field of expertise is transcriptional regulation using genomic techniques, however I am not familiar with the model *Schmidtea mediterranea*.Reviewer #3 (Evidence, reproducibility and clarity (Required)):Neiro et al. capitalize on existing genomic data for the planarian *Schmidtea mediterranea* and new ChIP-seq and ATAC-seq data to use computational approaches to identify putative enhancers in the planarian genome. They integrate analysis of enhancers with transcription factor binding sites to generate testable hypotheses for the regulatory function of transcription factors active in stem cells or control of cell lineage trajectories. Their work creates an excellent resource for future work to resolve the regulatory logic underpinning stem cell biology and tissue regeneration in planarians.

We are glad the reviewer likes our research.

Major: Overall, the work in this manuscript and methodology are well executed and presented. However, the authors should consider the following comments to improve the clarity and accessibility of the data and interpretations.1) The new transcriptome does not appear to be publically accessible. The links to Github resources are broken, and there is nothing on Neiro's Github page. Will the new transcriptome be integrated with Planmine?

The new *annotation* has been available for over a year as we wished the community to have access to it ASAP (see Garcia Castro, 2021, Genome Biology https://doi.org/10.1186/s13059-021-02302-5). We tested the links in the paper before depositing our preprint and after review and they seemed to work for us both within and outside our institutional network. We can only apologize if they were broken or have not worked for the reviewer. We are unclear if this new annotation will be included in Planmine, but we have asked the colleagues maintaining this database to consider including it. Our recent analyses of Planmine suggest that our annotation is now all included as all the new genes have appeared, even if our work is not explicitly referenced.

2) Figure 1: Ternary plot in 1F. The legend is not clear or could be explained better. What is the metric? It could be my misunderstanding, but I didn't consider the ternary plots as insightful or unnecessary. Perhaps the authors can expand on what they are showing.

These plots are important in demonstrating the distribution of mRNA expression of all genes across cell sorted compartments. Given the broad lineage relationship between sorted cell compartments This analysis allows us to identify genes expressed predominantly in one cell compartment or another, or across a specific transition. For example, genes enriched in Χ2 cells and Xins, but not X1 are likely to be enriched in post-mitotic differentiating progeny and differentiated cells. In contrast to single cell data where expression data can be sparse this analysis with bulk data allows identification and assignation of lowly expressed genes, like transcription factors. We will provide some further explanation of this in the revised text.

1I is a map of exons, not alternative splicing. So, it isn't clear what the authors intend t show. Are the specific exons that are more likely to be spliced? Is the figure necessary?

We wished to demonstrate the power of the annotation approach and the richness of the annotation for looking at alternate splicing. We have provided a more informative figure that indicates the variety of splice forms. We apologize for this oversight.

3) Figure 2: 2A labels Xins as irradiation responsive. Is this the case (just making sure)?

The reviewer is correct, this is wrong! This should read “insensitive” or “resistant” In Figure 1A. We thank the reviewer for spotting this error. We have fixed this.

2F-G: Ternary plot in F seems redundant with G, but that could be my lack of understanding. In 2G, what is represented on the plots on the right of the hierarchical clusters?

The ternary plot (2F) and heatmap of hierarchical clustering (2G) are complementary ways to visualize the proportional expression values of transcription factors. The ternary plot (2F) allows an overview of all the proportional expression values, while the heatmap (2G) shows how the proportional values may be grouped into clusters of similar expression profiles and displays the relative size of these clusters. For example, the heatmap shows that the clusters of X1 and Xins are more prominent than Χ2, suggesting that there are relatively a few Χ2-specific transcription factors. We have edited the text to better to explain this difference.

4) Figure 3: The heat maps need a legend (i.e., please define the colors). In addition, labeling the figures could help the reader. For example, in G-J, a header about the different experiments above each map, such as "enhancers" and "random," etc., would make the figure more accessible.

We agree the figures to be more easily interpretable and provided an independent scales and labels for the heatmaps.

5) Figure 5: Although it is in the figure legend, the authors could label the 6th track as "RNA-seq in X1."

We have added this to the figure.

6) Section 2.6 second page last sentence of the first paragraph "GRN of asexual reproduction is not active in neoblasts" data in the supplement? Is it not shown?

We apologize for this poorly written sentence. In line with Reviewer 2s comments this statement needs to be toned down and clarified. The raw information is included in the general table of enhancers (Supplementary File 2), but the genomic tracks visually highlighting the motifs at the promoters of lox5b and post2b are presented now Figure S4 A and B.

7) Discussion: The discussion about pluripotency factors in planarians could be expanded. The authors could contrast the study's findings with Önal et al. 2012.

We agree have expanded our discussion. But we very little is known with regard to TF roles in controlling pluripotency in adult stem cells.

Minor: The manuscript has no page numbers or line numbers, so I'll provide a general location of the potential issues.1) Section 2 – newly identified isoforms are shorter (1656 vs. 1618). Is the order of the median length reversed?

Yes, we have corrected this.

2) No mention of Figure S1B in the text.

We have added a correct reference to this figure in the presentation of transcript diversity.

3) Figure 1H should be 1I in the text?

Yes, we have corrected this

4) The discussion contains some minor typos and grammatical errors.

We have addressed with careful rereading.

We thank the reviewer for spotting these errors and we will fix them in revision.

Reviewer #3 (Significance (Required)):Neiro et al. provide an excellent resource for the planarian community. The paper is generally very well written and easy to read. The new transcriptome described, which improves the annotation of the planarian genome, should be made readily available. It would be excellent if the transcriptome could be incorporated in Planmine.

The annotation (without broad analysis) has been available since the pre-print for Garcia Castro, 2021, Genome Biology was deposited in BioRxiv.

Furthermore, the authors provide a comprehensive list of transcription factors in the planarian *Schmidtea mediterranea*. Their work provides insight into which factors are highly expressed in the stem cell compartment. Their computational identification of transcription factors and putative enhancers will be helpful to the growing community of researchers studying stem cell and regenerative biology using planarians. In addition, the large dataset generated in this study could inform studies in the evolution of regulatory sequences and transcription factor function.Referees cross-commentingThe data presented are well supported by previous studies. As noted by the authors, it is not possible to make transgenic planarians, and thus the field needs to rely on indirect methods. The authors focus on using the stem cell population, which can be isolated from the animals. Overall, I don't think additional experiments are necessary. Additional RNAi experiments combined with RNA-seq (using the stem cells) could take 6-12 months to complete. I believe this is a solid contribution that should be framed as a resource paper. The authors should pay close attention to Reviewer #2's suggestions and edit the paper accordingly.I have 20 years of experience in the field. It would be unreasonable to ask the authors to do more experiments, especially in this post-pandemic environment. I hope this helps.

We thank the reviewer for their comments.